# ComboBench: Can LLMs Manipulate Physical Devices to Play Virtual Reality Games?

## Abstract

Virtual Reality (VR) games require players to translate high-level semantic actions into precise device manipulations using controllers and head-mounted displays (HMDs). While humans intuitively perform this translation based on common sense and embodied understanding, whether Large Language Models (LLMs) can effectively replicate this ability remains underexplored. This paper introduces a benchmark, ComboBench, evaluating LLMs' capability to translate semantic actions into VR device manipulation sequences across 262 scenarios from four popular VR games: Half-Life: Alyx, Into the Radius, Moss: Book II, and Vivecraft. We evaluate twelve LLMs, including GPT-3.5, GPT-4, GPT-4o, GPT-5.1, Gemini-1.5-Pro, Gemini-3-Pro, Claude-Sonnet-4.5, Grok-4, GLM-4-Flash, LLaMA-3-8B, LLaMA-3-70B, and Mixtral-8x7B, compared against annotated ground truth and human performance. Our results reveal that while top-performing models like Gemini-3-Pro demonstrate strong task decomposition capabilities, they still struggle with procedural reasoning and spatial understanding compared to humans. Performance varies significantly across games, suggesting sensitivity to interaction complexity. Few-shot examples substantially improve performance, indicating potential for targeted enhancement of LLMs' VR manipulation capabilities. We release all materials at https://sites.google.com/view/combobench.

## 1 Introduction

Large Language Models (LLMs) have demonstrated remarkable proficiency in general-purpose task solving (Qin et al., 2023), conquering complex domains such as code (Lee et al., 2024; Lam et al., 2025) or math (Lu et al., 2024) problems. While they exhibit increasingly more human-like characteristics (Huang et al., 2024; Liang et al., 2023), an essential attribute of human intelligence is still underexplored: the ability to rapidly learn and apply unfamiliar concepts by leveraging common sense, prior experiences, and a repertoire of cognitive skills.

This is particularly evident in novel interactive environments like video games, where players quickly master device manipulations (atomic actions) and combine them to achieve complex semantic goals. Virtual Reality (VR) games elevate this challenge. They demand not only the execution of atomic actions via physical devices (*e.g.*, Head-Mounted Displays (HMDs) and controllers) but also the inference of complex, often uninstructed, semantic actions. For instance, in *Half-Life: Alyx* (Valve, 2020), when asked to "surrender," players might instinctively raise their controller-held hands even if not explicitly taught.

Such translation of high-level intent into a sequence of physical device manipulations engages a suite of cognitive abilities: (1) *Task decomposition*: Breaking down a high-level semantic action (*e.g.*, "tame the horse" and "plant wheat") into a coherent series of intermediate steps. (2) *Procedural reasoning*: Understanding the logical and temporal order of these steps, including prerequisite conditions or concurrent actions (*e.g.*, the need to till soil before planting seeds). (3) *Spatial reasoning & contextual awareness*: Interpreting instructions within a 3D spatial context (*e.g.*, "move HMD towards the Creeper" and "crouch through the gap") and understanding environmental cues or object states (*e.g.*, recognizing a door is open/closed and acting accordingly). (4) *Object interaction & tool use understanding*: Correctly mapping intended sub-actions to specific VR device manipulations (*e.g.*, knowing which button to press to "use" an item, and how to manipulate a controller to simulate "swinging" a tool like a pickaxe). This involves understanding the affordances of virtual objects

and tools. (5) *Motor action mapping & VR procedural transfer*: Translating abstract actions (*e.g.*, "press," "move," and "trigger") into specific, executable VR controller commands, potentially by adapting from provided examples or general knowledge of VR interaction paradigms. This touches upon a form of simulated embodied reasoning. (6) *Judgment of termination/continuation conditions*: Recognizing when a sub-task or a looped action is complete (*e.g.*, "mine until the block breaks" and "water until the plant grows"). Therefore, playing VR games serves as a rich testbed for evaluating if LLMs can bridge this gap between abstract understanding and grounded, physical interaction. Importantly, ComboBench is designed as a text-to-action benchmark: models receive only textual descriptions of high-level goals and must generate textual sequences of device manipulations. No visual or other multimodal inputs are provided, isolating the pure linguistic reasoning capability.

Virtual reality provides a distinctive testbed for evaluating embodied reasoning in large language models. While domains such as robotics and web agents also involve long-horizon decision-making, VR occupies a practical middle ground that foregrounds the core challenge of translating abstract linguistic intent into precise, physically grounded, and spatially coherent motor commands. Compared to physical robotics, VR enables complex, physics-based interactions without incurring real-world safety risks, hardware costs, or slow experimental cycles, thereby supporting rapid, scalable, and perfectly reproducible evaluation of embodied control. Relative to web or other digital agents that primarily operate over discrete symbolic actions (e.g., button clicks), VR requires reasoning in continuous 3D space, sensitivity to object affordances, and modeling of the temporal dynamics of manipulation, demanding a form of simulated embodiment not captured by many existing agent benchmarks. ComboBench is thus designed not as a generic long-horizon benchmark, but specifically to probe the interface where abstract knowledge must be realized as grounded physical action—a capability that is central to the development of general-purpose agents.

To systematically evaluate LLMs' ability to perform this crucial translation, we introduce ComboBench, which stands for Cognitive-Oriented Manipulation Benchmark for game combos using physical VR devices. It comprises 262 scenarios derived from four popular VR games: *Vivecraft* (Vivecraft, 2013) (Minecraft in VR), *Half-Life: Alyx* (Valve, 2020), *Moss: Book II* (Polyarc, 2022), and *Into the Radius* (CMGames, 2019). Each scenario presents a high-level semantic action, and the ground truth consists of a fine-grained sequence of VR device manipulations required to achieve it. These sequences are annotated by experienced VR players, allowing us to analyze LLM-generated outputs at the step-level and map their successes and failures to the aforementioned cognitive abilities. For example, failing to "press the X button" after "moving the HMD towards the Creeper" might indicate a lapse in procedural reasoning or object interaction understanding for that specific step.

We evaluate twelve LLMs, including GPT-3.5 (OpenAI, 2022), GPT-4 (OpenAI, 2023), GPT-4o (Hurst et al., 2024), GPT-5.1 (OpenAI, 2025), Gemini-1.5-Pro (Team et al., 2024), Gemini-3-Pro (Google, 2025), Claude-Sonnet-4.5 (Anthropic, 2025), Grok-4 (xAI, 2025), GLM-4-Flash (GLM et al., 2024), LLaMA-3-8B (Grattafiori et al., 2024), LLaMA-3-70B (Grattafiori et al., 2024), and Mixtral-8x7B (Jiang et al., 2023). We design a multi-dimensional scoring approach that assesses: (1) high-level semantic action understanding, (2) procedural step correctness, and (3) device-specific manipulation accuracy, allowing for fine-grained analysis of where each model succeeds or struggles in the translation process. Our findings reveal significant variation in model performance across cognitive capabilities. All models demonstrate strong task decomposition abilities but show pronounced weaknesses in motor action mapping and procedural reasoning. Gemini-3-Pro exhibits the most balanced performance across capabilities, while even advanced models like GPT-5.1 struggle with spatial reasoning compared to human performance. Few-shot examples substantially improve outcomes, particularly for procedural understanding, with diminishing returns beyond three examples. Performance also varies considerably across games, with models generally performing better in environments with more consistent interaction patterns (Vivecraft) than those requiring nuanced controller manipulations (Half-Life: Alyx). These results highlight specific cognitive gaps in current LLMs' ability to perform simulated embodied reasoning for VR interactions and identify targeted areas for improvement toward more capable virtual agents. Our contributions are:

- We introduce ComboBench, the first benchmark designed to evaluate LLMs' fine-grained cognitive abilities in translating high-level text semantic actions into text VR device manipulations, comprising 262 human-annotated scenarios from four diverse VR games.
- We define a set of key cognitive abilities crucial for VR interaction and design ComboBench to enable step-level analysis of LLM performance against these dimensions.

- We conduct a comprehensive evaluation of twelve state-of-the-art LLMs, providing a nuanced analysis of their strengths and weaknesses across these cognitive abilities and offering insights into the current frontiers of LLM-driven VR interaction.

## 2 COMBOBENCH DESIGN AND CURATION

### 2.1 COGNITIVE CAPABILITY TAXONOMY DEVELOPMENT

To ground our evaluation in cognitive theory, we collaborated with three experts in cognitive science and educational psychology who specialize in spatial cognition, procedural learning, and embodied interaction. Building on their feedback and an analysis of representative VR interaction scenarios from our benchmark, we converged on six core cognitive capabilities that are critical for translating semantic goals into VR device manipulations: (1) task decomposition, i.e., breaking high-level goals into sequentially ordered sub-tasks; (2) procedural reasoning, i.e., understanding causal relationships and temporal dependencies between actions; (3) spatial reasoning and contextual awareness, i.e., interpreting spatial layouts and environmental cues to guide action selection; (4) object interaction and tool-use understanding, i.e., inferring the affordances and functional properties of virtual objects; (5) motor action mapping and VR procedural transfer, i.e., mapping abstract action descriptions to concrete controller operations; and (6) judgment of termination and continuation conditions, i.e., recognizing when an action sequence has achieved its goal or requires repetition. These six dimensions form the taxonomy that underpins our subsequent analyses of LLM and human performance.

**Taxonomy Refinement.** Following the interviews, we synthesized the experts' insights through thematic analysis. Areas of consensus were directly incorporated into our taxonomy, while divergent perspectives were reconciled through follow-up consultations. This iterative process resulted in the identification of six core capability dimensions that comprehensively capture the cognitive demands of VR interaction: (1) Task decomposition: The ability to break down high-level goals into sequentially ordered sub-tasks. (2) Procedural reasoning: Understanding causal relationships between actions and their temporal dependencies. (3) Spatial reasoning & contextual Awareness: Processing spatial relationships and interpreting environmental cues for action selection. (4) Object interaction & tool use understanding: Comprehending affordances and functional properties of virtual objects. (5) Motor action mapping & VR procedural transfer: Translating conceptual actions into specific physical device manipulations. (6) Judgment of termination/continuation conditions: Recognizing completion states or conditions requiring repeated action.

### 2.2 GAME SELECTION CRITERIA AND PROCESS

To ensure a diverse and relevant set of VR interaction paradigms, we selected games based on a systematic process. First, we queried the Steam store (web, 2023) filtering for titles tagged as "VR Only" and available in "English," sorting the results by user review scores in descending order. We then iteratively examined games from this ranked list, focusing on their primary genre as categorized by Steam. To ensure genre diversity, we prioritized games from genres not yet represented in our collection. A crucial selection criterion was the availability of comprehensive textual walkthroughs. For each candidate game, we searched for detailed guides using keywords such as "walkthrough," "guide," or "tutorial." A walkthrough was deemed sufficiently detailed if it provided unambiguous, step-by-step instructions enabling the completion of core game objectives or specific complex tasks. Following this methodology, we selected four popular and critically acclaimed VR games representing distinct genres and interaction styles for ComboBench: (1) *Vivecraft* (Vivecraft, 2013) (Open-world sandbox, crafting) (2) *Half-Life: Alyx* (Valve, 2020) (First-person shooter, puzzle-solving, physics-based interaction) (3) *Moss: Book II* (Polyarc, 2022) (Third-person action-adventure, puzzle-platformer) (4) *Into the Radius* (CMGames, 2019) (First-person survival shooter, exploration) Such selection provides a rich variety of control schemes and task complexities for evaluating LLMs.

### 2.3 SCENARIO DEFINITION: SEMANTIC ACTION IDENTIFICATION

For all selected games, eight data annotators, comprising undergraduate and postgraduate computer science students with at least two years of programming experience and sufficient knowledge about VR games, manually identified salient semantic actions from the collected textual walkthroughs.

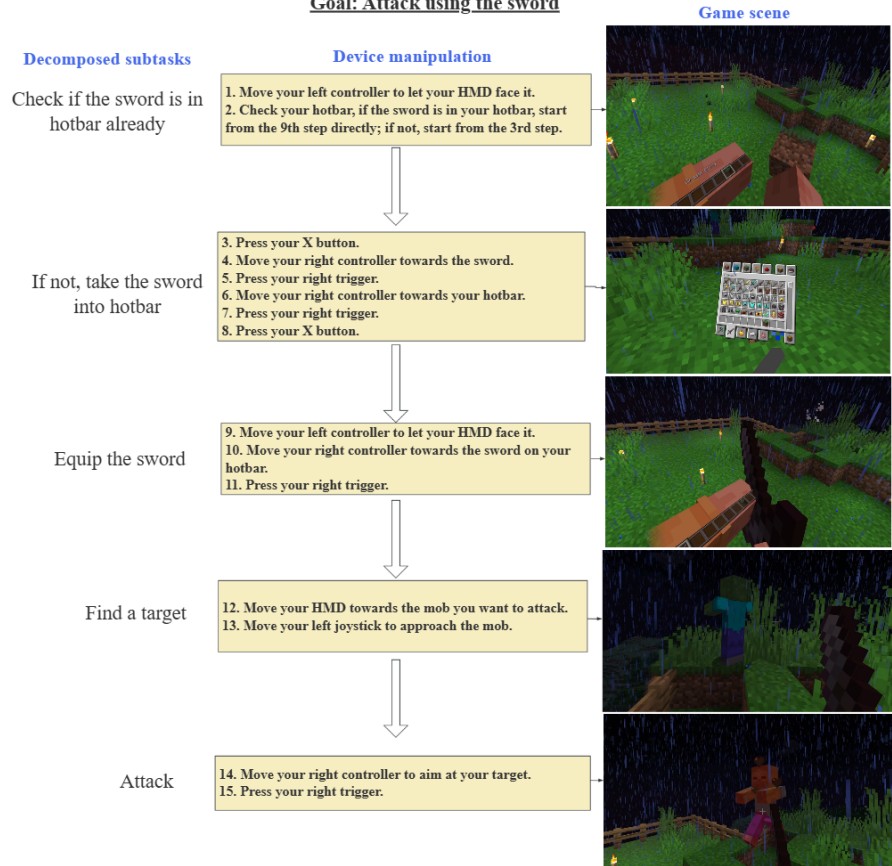

Figure 1: Device Manipulation Example for Task: "Attack using the sword" (*Vivecraft*)

Figure 2: Example of a Good vs. a Poor Model Generation for the "Attack using the sword"

Semantic actions were defined as high-level, goal-oriented tasks described in the walkthroughs (e.g., "tame the horse," "kill the creeper," "solve the gravity glove puzzle") that necessitate a sequence of fine-grained VR device manipulations to accomplish. We focused on scenarios that: (1) involve complex interactions not always explicitly detailed in in-game tutorials, (2) often constitute essential steps or objectives required for game progression. A concrete example is shown in Figure 1. This process resulted in the identification of 262 distinct scenarios across the four games.

## 2.4 ANNOTATION OF VR DEVICE MANIPULATIONS

Experienced VR users from our annotation team then played through each identified semantic action in the respective games using Oculus Quest 2 VR hardware. The objective was to record the precise sequence of device manipulations required to complete each semantic action. The annotation process captured the following details for each step within a manipulation sequence: (1) **Device used:** Specification of whether the HMD or a controller was used. (2) **Controller specificity:** If a controller

was used, and the action was hand-specific (e.g., primary hand for a tool), the annotation indicated whether the left or right controller was required. If either controller could perform the action, this was noted as "left or right controller." (3) **Operation type and parameters:** (i) *Movement:* For actions involving device movement (HMD or controller), the direction (e.g., "towards the Creeper," "upwards") or target position was recorded. (ii) *Button presses:* The specific button involved and the action (e.g., "press X button," "release trigger") were noted. (iii) *Joystick/thumbstick manipulation:* The direction of joystick push (e.g., "push left thumbstick forward") was recorded. (4) **Sequential composition:** For complex semantic actions composed of multiple, distinct sub-actions that might have been annotated individually, the sequence and composition of these simpler actions were explicitly recorded.

### 2.5 Cognitive Capability Labeling Using LLMs

A critical aspect of ComboBench is the annotation of each manipulation step with the specific cognitive capabilities it engages. This fine-grained labeling enables precise analysis of where LLMs succeed or fail in the VR interaction translation process. **(1) Initial Human Annotation.** To begin, our annotators manually labeled a subset of 50 manipulation sequences (approximately 20% of the dataset), assigning relevant capability categories to each step based on the taxonomy described in Section 2.1. For example, in the sequence required to "tame a horse" in Vivecraft, the step "equip the saddle by pressing the Y button while looking at the inventory slot containing the saddle" was labeled with "Object Interaction & Tool Use Understanding" and "Motor Action Mapping." **(2) LLM-Assisted Annotation Pipeline.** We then developed an LLM-assisted annotation pipeline to scale this process to the entire dataset. Specifically: ① We used the human-annotated examples as few-shot demonstrations for GPT-4o. ② For each unlabeled manipulation step, we provided the LLM with: [2.a] The semantic action context (e.g., "taming a horse in Vivecraft"). [2.b] The specific manipulation step to label. [2.c] The preceding and following steps (when available). [2.d] Detailed descriptions of each capability category. [2.e] Three few-shot examples with explanations of why each capability was assigned. ③ The LLM generated capability labels along with justifications for each assignment. ④ Human annotators reviewed the LLM-generated labels, making corrections when necessary. The review process revealed an 89.7% agreement rate between LLM-assigned labels and human judgments. **(3) Multi-label Distribution.** Most manipulation steps engaged multiple cognitive capabilities simultaneously. On average, each step was associated with 2.3 capability categories ($\sigma$ = 0.8). The most frequently co-occurring capabilities were "Motor Action Mapping" and "Object Interaction & Tool Use Understanding" (present together in 68% of steps), reflecting the inherent coupling between understanding virtual object affordances and translating this understanding into physical manipulations.

### 2.6 Contextualization and Verification

To further contextualize the annotated actions and aid in verification, we sourced or recorded gameplay videos corresponding to the textual walkthroughs for each game. For each annotated semantic action and its constituent manipulation steps, we recorded the corresponding timestamps in these videos. This allows for visual verification of the annotated sequences and provides richer context for understanding the actions. If suitable public gameplay videos matching the exact walkthrough steps were unavailable, our annotators recorded their own gameplay sessions while performing the actions.

## 3 Experiments

### 3.1 Model Selection

We evaluate twelve state-of-the-art LLMs spanning different model families and scales, including GPT-3.5, GPT-4, GPT-4o, GPT-5.1, Gemini-1.5-Pro, Gemini-3-Pro, Claude-Sonnet-4.5, Grok-4, LLaMA-3-8B, LLaMA-3-70B, Mixtral-8x7B, and GLM-4-Flash. This selection enables both cross-family comparisons and analysis of scaling effects within the same model family. We also perform human evaluation to validate the average human capabilities for comparison, when humans are given exactly the same input as LLMs. For all experiments, we used the official APIs for proprietary models and Hugging Face implementations for open-source models. Temperature was set to 0 across all

models to minimize non-deterministic outputs. For embedding calculations, we utilized OpenAI's text-embedding-3-large model via their API.

## 3.2 Evaluation Metrics

To comprehensively evaluate the capability of LLMs in translating semantic actions into VR device manipulations, we propose a multi-dimensional evaluation framework with four distinct metrics. These metrics collectively capture different aspects of model performance in ComboBench, ranging from strict matching to more flexible semantic alignment. The semantic similarity between predicted and ground-truth steps is computed using the cosine similarity of their sentence embeddings extracted from OpenAI's text-embedding-3-large model

**Strict Step-by-Step Matching (SSM).** Our first metric evaluates the exact matching between model-generated and ground truth steps, enforcing both sequence length equivalence and semantic alignment: $\text{SSM} = \frac{\text{Number of correctly predicted sequences}}{\text{Total number of sequences}}$. A sequence is considered correctly predicted only when: the number of steps in the generated sequence equals that of the ground truth, and every step in the generated sequence has a cosine similarity above a threshold of 0.8387 with its corresponding step in the ground truth. This strict metric serves as a measure of precision in reproducing exact device manipulation sequences and rewards models that can generate complete, step-accurate instructions. The threshold of 0.8387 was empirically determined by analyzing the cosine similarity distribution on a held-out set of human-paraphrased action steps. Specifically, we collected semantically equivalent but linguistically varied human annotations for 50 action steps and computed pairwise similarities. The threshold corresponds to the 5th percentile of similarities between these semantically equivalent pairs, ensuring that only highly confident matches are accepted while accommodating natural linguistic variation.

**Common Subsequence Evaluation.** We further introduce two complementary metrics based on common subsequence alignment to assess partial correctness: (1) **Normalized Step Alignment Score (NSAS)** This metric quantifies the alignment between the model-generated sequence and ground truth while accounting for missing and additional steps: $\text{NSAS} = \frac{(|C|-|M|-|A|)-\min_{\text{all\_samples}}}{|G|\cdot(\max_{\text{all\_samples}}-\min_{\text{all\_samples}})}$, where: $|C|$ represents the count of correctly matched steps in the common subsequence, $|M|$ represents missing steps from the ground truth, $|A|$ represents additional steps generated by the model, $|G|$ represents the total number of steps in the ground truth, $\min_{\text{all\_samples}}$ and $\max_{\text{all\_samples}}$ represent the minimum and maximum raw scores across all evaluations, enabling consistent normalization This score is normalized across the entire dataset to ensure fair comparison across different models and scenarios. (2) **Sequential Order Preservation (SOP)** The SOP metric specifically assesses the model's ability to maintain the correct procedural ordering of steps: $\text{SOP} = \frac{|\text{Steps correctly ordered and matched}|}{|G|}$. This metric evaluates whether the steps in the matched subsequence maintain their ordinal positions (e.g., step 1 followed by step 2, etc.) in both the ground truth and model output, capturing the model's procedural reasoning capabilities.

**Semantic Step Coverage (SSC).** Our final metric adopts a more flexible matching approach to evaluate semantic coverage of critical actions: $\text{SSC} = \frac{|\text{MR steps matched to any GT step}|}{|\text{MR}|}$, where a model result (MR) step is considered matched if it has a cosine similarity above the threshold (0.8387) with any step in the ground truth (GT). This metric computes the proportion of generated steps that semantically align with at least one ground truth step, regardless of position.

## 3.3 RQ1 & RQ3: LLM Performance Across VR Games

## 3.4 Experimental Results

We analyze and answer the following Research Questions (RQs): **(RQ1)** How do state-of-the-art LLMs perform in translating semantic actions into VR device manipulations across different VR games? **(RQ2)** How does the number of few-shot examples affect LLMs' ability to execute this translation? **(RQ3)** Do LLM and human performance exhibit significant variations across the four different VR games, potentially indicating sensitivity to game mechanics and interaction complexity? **(RQ4)** Which cognitive capabilities do current LLMs excel at, and where do they struggle? **(RQ5)** How do LLMs compare to human performance in VR device manipulation tasks?

Table 1: Overall performance comparison of LLMs across VR games (5-shot setting). Best model performance per metric is **bolded**, second best is underlined.

| Model | Half-Life: Alyx | | | | Into the Radius | | | | Moss: Book II | | | | Vivecraft | | | |
|---|---|---|---|---|---|---|---|---|---|---|---|---|---|---|---|---|
| | NSAS↑ | SOP↑ | F1$_{SOP}$↑ | SSC↑ | NSAS↑ | SOP↑ | F1$_{SOP}$↑ | SSC↑ | NSAS↑ | SOP↑ | F1$_{SOP}$↑ | SSC↑ | NSAS↑ | SOP↑ | F1$_{SOP}$↑ | SSC↑ |
| GPT-3.5 | 0.858 | 0.123 | 0.287 | 0.143 | 0.662 | 0.169 | 0.226 | 0.137 | 0.782 | 0.169 | 0.207 | 0.186 | 0.922 | 0.043 | 0.098 | 0.067 |
| GPT-4 | 0.853 | 0.125 | 0.258 | 0.172 | 0.693 | 0.189 | 0.328 | 0.177 | 0.824 | 0.218 | 0.336 | 0.220 | 0.927 | 0.137 | 0.437 | 0.081 |
| GPT-4o | 0.804 | 0.022 | 0.075 | 0.167 | 0.698 | 0.291 | 0.414 | 0.190 | 0.824 | **0.300** | 0.342 | 0.222 | 0.931 | 0.190 | **0.489** | 0.096 |
| GPT-5.1 | 0.903 | 0.251 | 0.320 | 0.493 | 0.857 | 0.062 | 0.172 | 0.269 | 0.888 | 0.206 | 0.300 | 0.383 | 0.864 | 0.109 | 0.221 | 0.144 |
| Gemini-1.5-Pro | 0.863 | 0.209 | 0.313 | 0.152 | 0.682 | 0.102 | 0.186 | 0.117 | 0.848 | 0.265 | 0.411 | 0.207 | 0.938 | 0.250 | 0.481 | 0.095 |
| Gemini-3-Pro | **0.929** | 0.309 | 0.427 | 0.650 | **0.927** | 0.280 | **0.478** | 0.611 | **0.928** | 0.262 | **0.487** | **0.572** | 0.895 | **0.379** | 0.343 | 0.228 |
| Claude-Sonnet-4.5 | 0.920 | 0.195 | 0.317 | 0.455 | 0.923 | 0.275 | 0.424 | **0.621** | 0.918 | 0.260 | 0.460 | 0.532 | 0.899 | 0.200 | 0.322 | 0.158 |
| Grok-4 | 0.924 | **0.351** | **0.430** | 0.655 | 0.911 | **0.320** | 0.396 | 0.564 | 0.918 | 0.231 | 0.428 | 0.558 | 0.869 | 0.270 | 0.319 | 0.311 |
| GLM-4-Flash | 0.836 | 0.076 | 0.183 | 0.149 | 0.618 | 0.096 | 0.186 | 0.149 | 0.749 | 0.087 | 0.174 | 0.165 | 0.909 | 0.000 | 0.045 | 0.061 |
| Mixtral-8x7B | 0.839 | 0.126 | 0.246 | 0.147 | 0.666 | 0.123 | 0.228 | 0.097 | 0.756 | 0.117 | 0.191 | 0.121 | 0.926 | 0.060 | 0.239 | 0.070 |
| LLaMA-3-8B | 0.848 | 0.126 | 0.279 | 0.162 | 0.644 | 0.242 | 0.317 | 0.168 | 0.823 | 0.283 | 0.349 | 0.200 | 0.929 | 0.039 | 0.122 | 0.042 |
| LLaMA-3-70B | 0.928 | 0.252 | 0.408 | **0.692** | 0.917 | 0.232 | 0.391 | 0.560 | 0.924 | 0.270 | 0.469 | 0.542 | 0.897 | 0.009 | 0.257 | **0.332** |
| Human | 0.845 | 0.090 | 0.240 | 0.110 | 0.684 | 0.148 | 0.257 | 0.181 | 0.817 | 0.112 | 0.328 | 0.174 | **0.935** | 0.122 | 0.482 | 0.084 |

Table 2: Overall performance across VR games and settings. We report the average scores for our four evaluation metrics: Strict Step-by-Step Matching (SSM), Normalized Step Alignment Score (NSAS), Sequential Order Preservation (SOP), and Semantic Step Coverage (SSC). Higher is better for all metrics. Bold indicates best model performance, underline indicates second best.

| Model | Average Across Settings | | | | Zero-Shot | | | | 5-Shot | | | |
|---|---|---|---|---|---|---|---|---|---|---|---|---|
| | SSM (%) | NSAS | SOP | SSC | SSM (%) | NSAS | SOP | SSC | SSM (%) | NSAS | SOP | SSC |
| GPT-3.5 | 1.4 | 0.781 | 0.063 | 0.066 | 0.8 | 0.771 | 0.003 | 0.046 | 2.1 | 0.791 | 0.128 | 0.095 |
| GPT-4 | 3.7 | 0.806 | 0.107 | 0.124 | 1.0 | 0.788 | 0.015 | 0.107 | 8.8 | 0.825 | 0.184 | 0.140 |
| GPT-4o | 5.3 | 0.797 | 0.138 | 0.141 | 0.6 | 0.785 | 0.015 | 0.108 | 10.9 | 0.806 | 0.228 | 0.161 |
| GPT-5.1 | 0.1 | 0.857 | 0.069 | 0.230 | 0.0 | 0.830 | 0.003 | 0.075 | 0.4 | 0.878 | 0.130 | 0.322 |
| Gemini-1.5-Pro | **5.8** | 0.813 | **0.146** | 0.142 | **2.1** | 0.795 | 0.010 | 0.124 | **11.7** | 0.832 | **0.236** | 0.162 |
| Gemini-3-Pro | 5.6 | **0.915** | 0.141 | **0.468** | 0.4 | **0.904** | 0.022 | **0.305** | 11.1 | **0.920** | 0.214 | 0.515 |
| Claude-Sonnet-4.5 | 4.5 | 0.906 | 0.115 | 0.340 | 0.0 | 0.890 | 0.004 | 0.115 | 8.4 | 0.915 | 0.182 | 0.442 |
| Grok-4 | 4.8 | 0.902 | 0.129 | 0.422 | 0.7 | 0.895 | 0.015 | 0.222 | 9.8 | 0.911 | 0.226 | 0.522 |
| GLM-4-Flash | 0.0 | 0.761 | 0.038 | 0.077 | 0.0 | 0.762 | 0.006 | 0.052 | 0.0 | 0.765 | 0.071 | 0.120 |
| Mixtral-8x7B | 1.1 | 0.784 | 0.068 | 0.079 | 0.0 | 0.777 | 0.002 | 0.040 | 2.2 | 0.796 | 0.105 | 0.107 |
| LLaMA-3-8B | 1.2 | 0.787 | 0.088 | 0.111 | 0.1 | 0.783 | 0.011 | 0.088 | 1.8 | 0.794 | 0.163 | 0.132 |
| LLaMA-3-70B | 3.8 | 0.909 | 0.126 | 0.409 | 0.2 | 0.898 | 0.003 | 0.160 | 8.5 | 0.916 | 0.191 | **0.531** |
| Human | 1.2 | 0.833 | 0.122 | 0.159 | – | – | – | – | – | – | – | – |

Tables 1, 2, and 3 present comprehensive performance metrics for all evaluated LLMs across the four VR games. Our analysis reveals substantial variations in model capabilities and game-specific challenges. Gemini-3-Pro emerges as the strongest performer overall, achieving the highest NSAS scores in three of the four games (Half-Life: Alyx: 0.929, Into the Radius: 0.927, Moss: Book II: 0.928), while maintaining competitive performance in Vivecraft (0.895). Grok-4 demonstrates particular strength in Half-Life: Alyx with the SOP score (0.351) and F1$_{SOP}$ (0.430), suggesting superior procedural reasoning capabilities in this specific game context.Claude-Sonnet-4.5 maintains consistently strong performance across all games, positioning itself as a reliable general-purpose model for VR interaction translation.

A striking pattern emerges in the SOP metrics, which vary dramatically across both models and games (0.000-0.379 range). While NSAS scores remain relatively high (mostly >0.75), indicating models can identify relevant steps, the low SOP values reveal fundamental difficulties in maintaining correct temporal ordering. This discrepancy is particularly pronounced in Vivecraft, where models achieve high NSAS scores (0.864-0.931) but struggle with step ordering (SOP: mostly below 0.200), suggesting that simpler interaction patterns may paradoxically lead to overconfidence in step sequencing.

Table 3: Cross-game performance variation (standard deviation across games) w/ 5-shot examples.

| Model | NSAS $\sigma$↓ | SOP $\sigma$↓ | F1$_{SOP}$ $\sigma$↓ | Game Gap↓ |
|---|---|---|---|---|
| GPT-3.5 | 0.110 | 0.061 | 0.084 | 0.085 |
| GPT-4 | 0.059 | 0.051 | 0.081 | 0.074 |
| GPT-4o | 0.068 | 0.137 | 0.184 | 0.127 |
| GPT-5.1 | 0.018 | 0.102 | 0.097 | 0.073 |
| Gemini-1.5-Pro | 0.099 | 0.093 | 0.127 | 0.095 |
| Gemini-3-Pro | 0.014 | 0.123 | 0.106 | 0.081 |
| Claude-Sonnet-4.5 | **0.009** | 0.110 | 0.134 | 0.084 |
| Grok-4 | 0.013 | 0.137 | 0.065 | 0.072 |
| GLM-4-Flash | 0.135 | 0.049 | 0.069 | 0.084 |
| Mixtral-8x7B | 0.114 | 0.031 | **0.065** | 0.070 |
| LLaMA-3-8B | 0.112 | 0.103 | 0.120 | 0.113 |
| LLaMA-3-70B | 0.012 | 0.106 | 0.077 | **0.065** |
| Human | 0.105 | **0.029** | 0.117 | 0.084 |

Analysis of performance variations (Table 3) reveals significant game-dependent effects. Vivecraft exhibits the highest average performance across models (0.864-0.931), likely due to its consistent block-based interaction paradigm inherited from Minecraft. In contrast, Into the Radius presents the greatest challenge, with notably lower NSAS scores (0.618-0.927) and high performance variance. This pattern suggests that games featuring realistic physics simulations and complex inventory management pose particular difficulties for current LLMs.

Interestingly, different models exhibit distinct strengths across game types. Grok-4 shows remarkable adaptability in Half-Life: Alyx compared to other models, while struggling in Vivecraft (NSAS 0.869). Gemini-3-Pro maintains the most balanced performance profile across games (Game Gap: 0.081), suggesting more robust generalization capabilities. Smaller models like Mixtral-8x7B and GLM-4-flash show disproportionate performance degradation in complex environments, with GLM-4-flash achieving zero SOP in Vivecraft despite reasonable NSAS scores. The substantial performance variations across games highlight the impact of interaction design on LLM capabilities. Games with discrete, well-defined actions (Vivecraft) enable higher model performance, while those requiring nuanced controller manipulation and spatial reasoning (Half-Life: Alyx, Into the Radius) expose current limitations. The correlation between game complexity and performance degradation is non-linear, moderate complexity (Moss: Book II) sometimes yields better results than simpler environments, suggesting that models may benefit from richer contextual cues in certain scenarios.

These findings collectively demonstrate that while state-of-the-art LLMs have made significant progress in understanding VR interactions, their performance remains highly sensitive to specific game mechanics and interaction paradigms. The gap between high NSAS scores and low SOP values across all games indicates that current models can identify relevant actions but struggle with the procedural reasoning required to sequence them correctly, which is an important capability for successful VR interaction.

Table 2 demonstrates that few-shot examples substantially improve LLM performance in VR device manipulation tasks, with the most dramatic gains observed in Sequential Order Preservation (SOP), where scores increase by 10–20x from near-zero baselines. All models benefit from in-context examples, though with diminishing

Table 4: Multi-Path Step Matching (MP-SSM) NSAS Score for Selected Actions

| Model | Average NSAS | Zero-Shot NSAS | 5-Shot NSAS |
|---|---|---|---|
| GPT-3.5 | 0.927 (↑8.3%) | 0.904 (↑4.3%) | 0.942 (↑9.7%) |
| GPT-4 | 0.949 (↑8.7%) | 0.937 (↑9.1%) | 0.955 (↑6.9%) |
| GPT-4o | 0.949 (↑11.5%) | 0.942 (↑13.7%) | 0.955 (↑9.3%) |
| Gemini-1.5-Pro | 0.939 (↑6.8%) | 0.923 (↑8.3%) | 0.947 (↑6.0%) |
| GLM-4-Flash | 0.926 (↑9.7%) | 0.896 (↑5.2%) | 0.939 (↑10.2%) |
| Mixtral-8x7B | 0.925 (↑7.6%) | 0.894 (↑3.0%) | 0.934 (↑9.2%) |
| LLaMA-3-8B | 0.940 (↑9.1%) | 0.926 (↑7.3%) | 0.940 (↑8.6%) |
| Human | 0.931 (↑4.5%) | – | – |

returns, the improvement from zero-shot to 3-shot (average NSAS gain: 2.1%, SOP: 10-fold increase) significantly exceeds that from 3-shot to 5-shot (NSAS: 1.4%, SOP: 20-50% relative gain). Gemini-3-Pro exhibits the strongest adaptability, achieving the highest 5-shot performance (NSAS: 0.920), while maintaining consistent improvements across all metrics. The differential impact across metrics reveals that few-shot examples primarily address procedural sequencing challenges (massive SOP improvements) more effectively than exact step matching (modest SSM gains), suggesting that demonstrations help models understand temporal dependencies in VR interactions but do not fully resolve the complexity of translating semantic actions into precise device manipulations.

**Scaling effects within model families**. The inclusion of LLaMA-3-70B alongside LLaMA-3-8B enables direct analysis of scaling effects within the same architecture. Table 2 shows that scaling from 8B to 70B parameters yields substantial improvements: average NSAS increases from 0.787 to 0.909, SSM from 1.2% to 3.8%, and SSC from 0.111 to 0.409. These gains are consistent across games, with the most pronounced improvements in semantic step coverage. LLaMA-3-70B achieves competitive performance with proprietary models like GPT-4o and approaches Gemini-3-Pro on several metrics, suggesting that open-source models can match proprietary systems when appropriately scaled.

### 3.5 RQ4: COGNITIVE CAPABILITIES ANALYSIS

We analyzed model performance across six cognitive capabilities required for effective VR interaction (Figure 3). By mapping evaluation metrics to capability scores (0-10 scale), we identified specific strengths and limitations in how LLMs approach spatial-mechanical reasoning tasks.

**Areas of strength:** All evaluated LLMs demonstrate strong task decomposition capabilities (7.8-8.5), with minimal performance gap compared to humans (8.2). Gemini-1.5-Pro leads with a score of 8.5, while even smaller models like Mixtral-8x7B (8.0) and GLM-4-flash (7.8) perform admirably. This suggests that segmenting high-level actions into component steps aligns well with the sequential reasoning abilities developed during language model pre-training.

**Areas of weakness:** Motor action mapping emerges as the most significant challenge (0.5-4.5), with all models struggling to precisely translate abstract actions into specific VR control manipulations. GPT-4o performs best in this dimension (4.5), but still falls short of robust capability. Procedural reasoning also shows substantial variation (2.3-7.0), with only Gemini-1.5-Pro approaching adequate

performance. Judgment of termination conditions represents another challenge area, with most models scoring below 5.0 (except Gemini-1.5-Pro at 6.0), compared to human performance (6.5).

**Model comparison:** Gemini-1.5-Pro demonstrates the most balanced performance profile, consistently outperforming other models in procedural reasoning (7.0), spatial reasoning (7.5), and termination judgment (6.0). GPT-4 variants show strong task decomposition and object interaction (5.3-5.7) but lag in procedural sequencing. LLaMA-3-8B shows surprisingly competitive performance in procedural reasoning (5.7), outperforming larger models like GPT-3.5-Turbo (4.3), suggesting architecture differences may be as important as scale.

### 3.6 RQ5: Comparison with Human

To contextualize our findings, we compare LLM performance against human baselines across our evaluation metrics. As shown in Tables 1 and 3, state-of-the-art LLMs demonstrate competitive performance with humans on several key dimensions. Human performance forms a strong but no longer dominant baseline when compared to state-of-the-art LLMs on our text-to-action translation task.

We note that our human baseline measures performance on the text-to-action-sequence translation task specifically, that is, humans were asked to write down step-by-step device manipulations given a semantic goal description, mirroring exactly the task given to LLMs. This setup ensures an apples-to-apples comparison but differs from in-situ VR performance, which would additionally involve real-time problem-solving, exploration, and motor execution. The surprising finding that models outperform humans on certain metrics (e.g., SOP in Half-Life: Alyx) thus reflects the difficulty of recalling and accurately sequencing complex interactions from memory, validating the challenge posed by our benchmark.

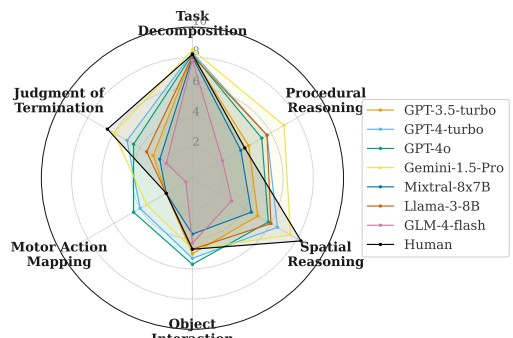

Figure 3: Cognitive capabilities of LLMs and humans in translating semantic actions to VR device manipulations. Higher scores (0-10 scale) indicate stronger abilities.

Across all four VR games, humans consistently achieve mid-to-high NSAS scores (0.684–0.935), indicating reliable identification of relevant steps, yet they are outperformed by the reasoning models in games, with systems like Gemini-3-Pro, Grok-4, and LLaMA-3-70B reaching NSAS values above 0.90 in most settings. More strikingly, humans lag behind top-performing models on SOP and SSC: while human SOP remains below 0.15 and SSC below 0.19 across games, models such as Grok-4, Gemini-3-Pro, and Claude-Sonnet-4.5 attain substantially higher procedural ordering and semantic coverage, often exceeding 0.30–0.65 on these metrics. These results suggest that, for the specific task of translating high-level VR goals into textual device-manipulation sequences, current LLMs not only match but frequently surpass human participants in both the completeness and the temporal structuring of the generated action steps.

Analysis of performance variance across games (Table 3) reveals striking similarities between human and high-performing model behavior. The standard deviation of human performance (0.084) closely aligns with that of Grok-4 (0.081) and Claude-Sonnet-4.5 (0.084), suggesting that both humans and advanced LLMs exhibit similar sensitivity patterns to game-specific interaction complexities. This convergence is particularly evident in structured environments like Vivecraft, where the consistency gap between humans and LLMs has substantially narrowed. Figure 3 illustrates the capability-wise performance comparison, revealing critical gaps in embodied reasoning. Humans maintain superior performance in spatial reasoning (8.3 vs. 7.5 for Gemini-1.5-Pro) and judgment of termination conditions (6.5 vs. 6.0). These differences are statistically significant ($p < 0.05$, Wilcoxon signed-rank test) and persist across all evaluated models. This performance gap suggests that while LLMs have achieved remarkable progress in understanding VR interaction semantics, they lack the grounded physical intuition that humans naturally apply when reasoning about three-dimensional manipulations and determining action completion states.

The convergence of human and LLM performance on certain metrics, coupled with persistent gaps in spatial and termination reasoning, indicates that current language models can effectively decompose VR tasks but struggle with aspects requiring embodied experience. This finding has important

implications for the development of future VR-capable AI systems, suggesting the need for training paradigms that better incorporate spatial and physical reasoning capabilities.

Table 4 presents the results of our Multi-Path Step Matching (MP-SSM) evaluation, in which each model and the human baseline are assessed against multiple distinct valid ground-truth solutions for each scenario. Compared to previous single-path assessments, all systems show substantial gains in NSAS scores, confirming that accounting for the diversity of human-authored action sequences provides a more accurate picture of their abilities. The human baseline, in particular, sees a marked improvement, with an average NSAS of 0.931 and a relative increase of 14.5%, establishing a meaningful reference for achievable performance. While all models register higher scores, the relative ranking among them is preserved: GPT-4 and GPT-4o lead overall, closely followed by LLaMA-3, Gemini-1.5, and GPT-3.5, with Mixtral and GLM-4 slightly behind. These findings reinforce that state-of-the-art LLMs remain competitive with expert humans even under this more realistic multi-solution evaluation, and highlight the importance of considering multiple valid approaches when measuring success in open-ended procedural tasks.

## 4  RELATED WORK

Recent work has explored LLMs as generalist agents for embodied reasoning. In robotics, *Say-Can* (Ahn et al., 2022) and *PaLM-E* (Driess et al., 2023) combine LLMs with affordance-based skill models or multimodal inputs to plan and execute actions, demonstrating that LLMs can decompose high-level goals into actionable steps when grounded in sensory input. Similar capabilities appear in virtual domains through agents like *Voyager* (Wang et al., 2023) and platforms like *MineDojo* (Fan et al., 2022), which showcase autonomous skill acquisition via code generation. However, these systems focus on code-level or symbolic outputs rather than physical device manipulation or spatially grounded motor control required in VR. Task decomposition has been studied via prompting strategies such as Chain-of-Thought (Wei et al., 2022) and ReAct (Yao et al., 2022), which improve multi-step planning coherence. LLMs generate structured action sequences in domains like household tasks (Shridhar et al., 2020) and scientific procedures (Wang et al., 2022), while code-as-policy paradigms (Liang et al., 2022) enable conditional and iterative actions through executable policy code. These approaches, however, often abstract away physical or spatial execution complexity. Several benchmarks assess grounded reasoning in interactive settings. Animal-AI (Mecattaf et al., 2024) evaluates embodied cognition through physics-based tasks, while ALFWorld (Shridhar et al., 2021), ScienceWorld (Wang et al., 2022), and MacGyver-style tasks (Tian et al., 2024) test instruction-following and object-use innovation, revealing LLMs' limitations in spatial reasoning and tool-use generalization. Concurrently, capability-oriented embodied evaluations have emerged: Embodied-Bench (Yang et al., 2025) unifies tasks with fine-grained error taxonomies; VLABench (Zhang et al., 2024a) targets long-horizon manipulation; EAI (Li et al., 2025) standardizes step-level diagnostics. GUI/OS/mobile benchmarks including OSWorld (Xie et al., 2024), SPA-Bench (Zhang et al., 2024b), WebArena (Zhou et al., 2023), Mind2Web (Deng et al., 2023), AndroidEnv (Toyama et al., 2021), and AppAgent/AppAgent v2 (Zhang et al., 2023; Li et al., 2024b) evaluate precise device interactions. On the robotics side, VLA policies such as RT-1/RT-2 (Brohan et al., 2022; 2023) and OpenVLA (Kim et al., 2024) map observations to actions, while large-scale 3D suites like Habitat 2.0/HAB (Savva et al., 2021), BEHAVIOR-1K/OmniGibson (Li et al., 2024a), and CALVIN (Mees et al., 2021) stress long-horizon rearrangement.

In contrast, ComboBench targets the translation of semantic goals into fine-grained, physically grounded VR device manipulations, enabling precise step-level analysis of embodied cognitive abilities critical for real-world interaction.

## 5  CONCLUSION

We introduced ComboBench, a benchmark evaluating LLMs' ability to translate semantic actions into VR device manipulations across 262 scenarios from four VR games. Our evaluation of twelve LLMs reveals that while models demonstrate strong task decomposition, they struggle with procedural reasoning and motor action mapping. Few-shot examples substantially improve performance, but significant gaps remain compared to human capabilities, highlighting the need for multimodal training approaches that incorporate spatial and embodied reasoning.

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

## A    PRELIMINARIES ON VIRTUAL REALITY

Virtual Reality (VR) represents a fundamentally distinct paradigm of human-computer interaction that transcends traditional interface boundaries. Unlike conventional computing systems that rely on indirect manipulation through keyboards, mice, and two-dimensional displays, VR creates immersive digital environments where users experience presence and embodiment. This paradigm shift necessitates a comprehensive understanding of both the technological infrastructure and the cognitive demands placed on users who must translate abstract intentions into concrete physical manipulations within virtual spaces.

The evolution of VR technology has progressed through several generations, from early tethered systems requiring substantial computational infrastructure to modern standalone devices that integrate processing, display, and tracking capabilities within compact form factors. Contemporary VR systems can be broadly categorized into three architectural approaches. PC-tethered headsets leverage external computational resources to deliver high-fidelity experiences with complex graphics and physics simulations. Standalone headsets, exemplified by devices like the Meta Quest series, incorporate integrated processors that balance performance with portability. Mobile-phone-based solutions represent an accessible entry point, utilizing smartphones as both display and processor, though with inherent limitations in tracking precision and computational capability.

The core hardware components enabling VR interaction form an integrated ecosystem of sensory input and output devices. Head-Mounted Displays (HMDs) serve as the primary visual interface, providing stereoscopic rendering that creates depth perception while simultaneously tracking head orientation and position through integrated sensors. This tracking enables natural viewing behaviors where users can examine virtual objects by physically moving their heads, mirroring real-world visual exploration patterns. Motion controllers, typically deployed in pairs to represent both hands, enable direct manipulation of virtual objects through a combination of positional tracking, button inputs, trigger mechanisms, and thumbstick controls. These devices must balance ergonomic considerations with functional complexity, providing sufficient input channels while maintaining intuitive operation. Spatial tracking systems, whether implemented through external sensors (outside-in tracking) or integrated cameras (inside-out tracking), monitor user movements with six degrees of freedom, capturing both translational and rotational motion to enable natural locomotion and interaction within virtual environments.

The ongoing evolution of VR hardware continues to introduce novel interaction modalities. Haptic gloves promise to deliver tactile feedback through actuators that simulate texture, resistance, and temperature. Full-body tracking systems capture skeletal motion to enable more nuanced avatar control and gesture recognition. Specialized peripherals, from steering wheels for racing simulations to weapon replicas for combat games, demonstrate the trend toward application-specific controllers that enhance immersion through physical affordances that match virtual interactions.

### A.1    INTERACTION PARADIGMS AND DESIGN PRINCIPLES

The design of VR interaction paradigms represents a delicate balance between leveraging users' existing motor skills and introducing novel control schemes that exploit the unique capabilities of virtual environments. Direct manipulation forms the foundation of most VR interactions, where users employ hand controllers to simulate natural actions like grasping, throwing, and pushing. This approach capitalizes on users' lifetime of experience with physical object manipulation but requires careful calibration of virtual physics to match expectations. The mapping between controller inputs and virtual hand movements must account for the absence of tactile feedback, often employing visual or auditory cues to confirm successful interactions.

Ray-casting emerged as an elegant solution to the fundamental challenge of interacting with objects beyond physical reach. By projecting virtual rays from controllers, users can select, manipulate, and activate distant objects without locomotion. This technique exemplifies how VR interaction design often augments natural human capabilities rather than strictly simulating physical constraints. Advanced ray-casting implementations incorporate features like ray curvature for improved ergonomics, variable ray length based on context, and visual feedback mechanisms that indicate interaction possibilities.

Gesture recognition systems interpret temporal patterns of controller or hand movement as discrete commands, enabling a rich vocabulary of interactions without relying on button combinations. These systems must balance recognition accuracy with user comfort, avoiding gestures that cause fatigue or require precise movements difficult to perform consistently. Machine learning approaches have enhanced gesture recognition capabilities, allowing for more natural and varied input patterns while maintaining reliable detection rates.

Symbolic input mechanisms address scenarios where direct physical analogues are impractical or inefficient. Virtual keyboards present unique challenges in VR, as users lack tactile feedback and must rely on visual confirmation of key presses. Solutions range from laser-pointer selection of virtual keys to gesture-based text entry systems that map hand movements to characters. Voice commands offer an alternative input modality that bypasses manual interaction entirely, though they introduce considerations around recognition accuracy, latency, and social acceptability in shared spaces.

## A.2 DEVELOPMENT PLATFORMS AND TECHNICAL CONSIDERATIONS

The creation of VR applications relies on sophisticated development ecosystems that abstract hardware complexity while providing fine-grained control over interaction mechanics. Unity and Unreal Engine have emerged as dominant platforms, offering comprehensive toolsets that handle rendering pipelines, physics simulation, spatial audio, and cross-platform deployment. These engines provide specialized VR interaction frameworks that standardize common patterns like object grabbing, teleportation, and menu systems, significantly reducing development complexity.

Hardware software development kits (SDKs) serve as the bridge between high-level application logic and device-specific capabilities. Meta's OpenXR initiative represents an industry effort to standardize VR/AR interfaces, enabling applications to target multiple hardware platforms without extensive modifications. Platform-specific SDKs like SteamVR and Oculus SDK continue to play important roles, offering access to proprietary features and optimizations that enhance performance on particular hardware.

Technical constraints fundamentally shape VR interaction design decisions. Maintaining consistent frame rates above 72Hz (and preferably 90Hz or higher) prevents motion sickness and ensures responsive interactions. This performance requirement influences every aspect of application design, from polygon counts and texture resolution to the complexity of physics simulations. Tracking precision varies across hardware platforms and environmental conditions, necessitating interaction designs that accommodate occasional tracking losses or reduced accuracy. Developers must also consider the diverse computational capabilities across the VR ecosystem, implementing scalable solutions that provide acceptable experiences on entry-level hardware while leveraging the capabilities of high-end systems.

## A.3 CHALLENGES IN VR INTERACTION

Despite remarkable technological progress, VR interaction continues to face fundamental challenges that impact user experience and limit application domains. The locomotion problem exemplifies the tension between physical and virtual spaces. While users may explore vast virtual environments, they remain constrained by finite physical play areas. Teleportation offers a practical solution but breaks immersion and can cause spatial disorientation. Artificial locomotion through thumbstick control risks motion sickness in susceptible users. More exotic solutions like omnidirectional treadmills or redirected walking techniques remain impractical for consumer applications.

The absence of comprehensive haptic feedback represents perhaps the most significant limitation in current VR systems. While controllers provide basic vibration feedback, they cannot simulate the rich tactile experiences of real-world interaction: the weight of objects, surface textures, temperature variations, or resistance to movement. This sensory gap creates a fundamental disconnect between visual expectations and physical sensations, requiring users to adapt their interaction strategies and often leading to reduced precision in manipulation tasks.

Interaction discoverability poses ongoing challenges as VR applications lack standardized interface conventions comparable to desktop or mobile platforms. Users encountering new VR experiences must often learn application-specific control schemes, gesture sets, and interaction patterns. The absence of persistent visual UI elements (to maintain immersion) exacerbates this challenge, as users

cannot easily reference control schemes during gameplay. This lack of standardization increases cognitive load and creates barriers to entry for new users.

Precision manipulation tasks highlight the limitations of current tracking systems and input devices. Tasks requiring fine motor control, such as threading a virtual needle or manipulating small components, prove challenging due to tracking jitter, lack of physical surfaces for hand stabilization, and absence of tactile confirmation. These limitations restrict the types of applications suitable for VR and influence interaction design toward larger, more forgiving target sizes and simplified manipulation schemes.

**(a) NSAS**

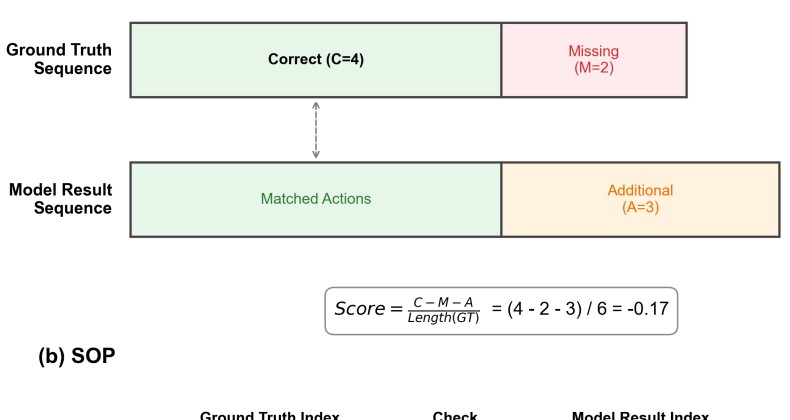

$$Score = \frac{C - M - A}{Length(GT)} = (4 - 2 - 3) / 6 = -0.17$$

**(b) SOP**

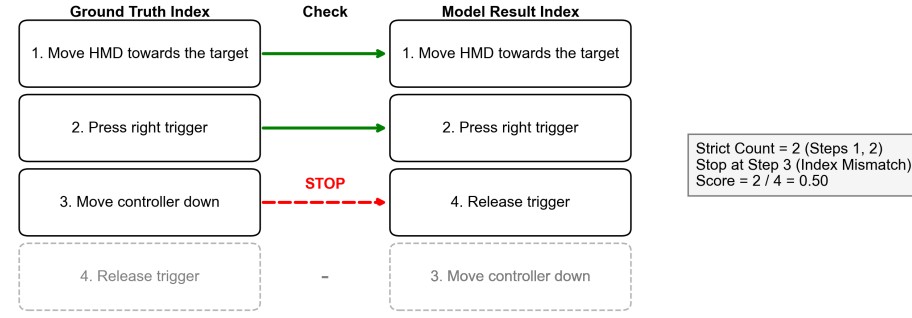

Figure 4: Overview of Strict Sequential Order Preservation (SOP) and Normalized Step Alignment Score (NSAS) Calculation

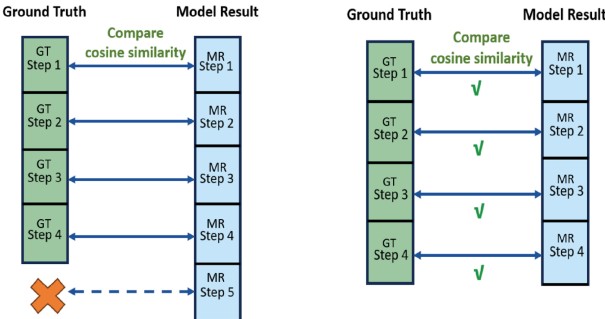

Figure 5: Overview of Strict Step-by-Step Matching (SSM) Calculation

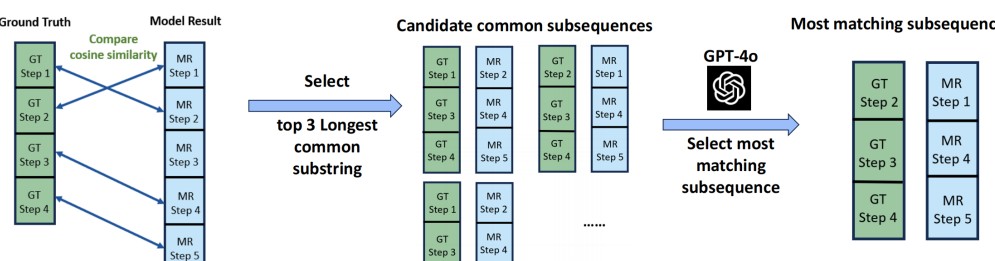

Figure 6: Overview of Common Subsequence Evaluation

# B EXPERT INTERVIEW.

To derive the taxonomy in Section 1.1, we conducted semi-structured interviews with three domain experts who specialize in cognitive science and educational psychology, with research backgrounds in spatial cognition, procedural learning, and embodied interaction. Each expert participated in a 90-minute online interview focused on identifying the cognitive abilities required to translate semantic goals into physical actions in virtual environments.

The interview protocol followed three phases: (1) an open-ended discussion of cognitive processes involved in VR interaction, (2) a structured review of preliminary capability categories synthesized from prior work, and (3) targeted refinement, where experts proposed additions, merges, and clarifications to these categories. We performed a thematic analysis over the interview transcripts to extract points of agreement and disagreement. Areas of consensus were directly incorporated into the taxonomy, while divergent views were reconciled through follow-up email consultations. This process yielded the six core capability dimensions reported in the main text.

# C EXPLANATION OF EVALUATION METRICS

## C.1 STRICT STEP-BY-STEP MATCHING (SSM)

Figure 5 illustrates the Strict Step-by-Step Matching (SSM) calculation process. SSM represents our most stringent evaluation metric, requiring exact correspondence between model-generated sequences and ground truth annotations. The calculation process operates as follows:

In the left panel, we observe a scenario where the ground truth contains 4 steps while the model result contains 5 steps. For SSM to register a match, two conditions must be satisfied: (1) the number of steps must be identical between ground truth and model output, and (2) each step must have a cosine similarity score above our threshold of 0.8387 with its corresponding ground truth step. In this example, the length mismatch alone disqualifies the sequence from being counted as correct, resulting in an SSM score of 0. The orange X symbol on the fifth model step visually indicates this length mismatch failure.

The right panel demonstrates a successful SSM match where both sequences contain 4 steps. Each model step is compared with its corresponding ground truth step using cosine similarity of their text embeddings. The green checkmarks indicate that all four step pairs exceed the similarity threshold, resulting in a successful match and contributing 1 to the SSM score. This metric's strictness explains why even high-performing models achieve relatively low SSM scores—any deviation in sequence length or individual step similarity results in complete failure for that sequence.

## C.2 COMMON SUBSEQUENCE EVALUATION

Figure 6 details our Common Subsequence Evaluation approach, which underlies the Normalized Step Alignment Score (NSAS) and Sequential Order Preservation (SOP) metrics. This evaluation method provides more nuanced assessment than SSM by identifying partial matches and preserved ordering within sequences.

The process begins with comparing each step in the ground truth and model result sequences using cosine similarity, as shown by the crossing blue lines in the leftmost panel. Unlike SSM's strict position-based matching, this approach allows steps to match regardless of their positions in the sequences. The algorithm then identifies the top 3 longest common subsequences where matched steps maintain their relative ordering.

In the example shown, multiple candidate subsequences are generated, each representing different ways steps from both sequences can be aligned while preserving order. The model (shown as GPT-4o) then selects the most matching subsequence based on the highest cumulative similarity scores. The final selected subsequence shows GT Steps 2, 3, and 4 matching with MR Steps 1, 4, and 5 respectively. This flexible matching approach allows the metrics to capture semantic correctness even when models include additional steps or present steps in slightly different positions.

The NSAS metric is calculated by considering the correctly matched steps (|C|), missing steps from ground truth (|M|), and additional steps in the model output (|A|), normalized by the total ground truth steps and scaled across the dataset. The SOP metric specifically evaluates whether matched steps maintain their sequential order, providing insight into the model's procedural reasoning capabilities.

### C.3 Cognitive Capability Score Derivation

The radar chart in Figure 3 presents normalized capability scores (0–10 scale) derived by aggregating model performance on scenario subsets that heavily engage each cognitive dimension. Each scenario in ComboBench was labeled with primary cognitive requirements during the annotation process described in Section 1.5. For example, scenarios labeled with high "Spatial Reasoning" complexity (e.g., "crouch through the gap," "navigate around the obstacle") form the subset used to compute spatial reasoning scores.

For each capability dimension $c$, the score for model $m$ is computed as:

$$\text{Score}_c^m = 10 \times \frac{\text{NSAS}_c^m - \min_{\text{models}}(\text{NSAS}_c)}{\max_{\text{models}}(\text{NSAS}_c) - \min_{\text{models}}(\text{NSAS}_c)}. \tag{1}$$

Here, $\text{NSAS}_c^m$ is the average NSAS score of model $m$ on scenarios primarily requiring capability $c$. This normalization ensures comparability across dimensions with different baseline difficulties.

# D  DETAILED PROMPT

**Prompt for Vivecraft Action Decomposition**

You are an expert VR game player deeply immersed in a VR game called Vivecraft. You are holding your VR controllers in both hands and view the game scene through your HMD. Your task is to thoroughly describe how you perform semantic actions in Vivecraft by breaking them down into step-by-step sequences of device manipulations in the given JSON format. Use precise and clear instructions, and include all necessary steps to ensure accurate execution of the action. The output should include atomic game actions and corresponding VR device manipulations. Your output must strictly follow this JSON format:

```
{
    "action ID": "<action ID>",
    "semantic action": "<action>",
    "atomic game action": [
        "1. <step_1>",
        ......
        "n. <step_n>"
    ],
    "device manipulations": [
        "1. <step_1>",
        ......
        "n. <step_n>"
    ]
}
```

Here is the introduction to the VR game Vivecraft to help you better decompose the atomic action and device manipulation:

`<game_intro/>`

Vivecraft is the mod that transforms Minecraft into an exceptional VR experience in room-scale or seated play. It is a sandbox game that allows players to explore, create, and survive in a blocky, procedurally generated world. The key mechanism of the game is listed below: – Mining and Crafting: Gather materials from the environment and craft tools, weapons, and other items using a crafting table. – Building: Use blocks to construct buildings, machines, and other structures. – Exploration: Discover various biomes with unique landscapes, resources, and mobs. – Combat: Defend against hostile mobs like zombies, skeletons, and creepers. – Farming and Animal Husbandry: Grow crops and breed animals for long-term survival.

`</game_intro>` Here is the general VR controller user guide that helps you to decompose the semantic action into device manipulation:

`<vr_device_guide/>`

- **HMD:** Provides immersive visual and auditory VR experience; displays 360-degree environments and delivers spatial audio.

- **Triggers (controllers):** Used for precise actions, including pressing virtual buttons and selecting or interacting with objects.

- **Grips (controllers):** Used for grabbing and manipulating objects, including grabbing, moving, rotating, and resizing; pressing the grip forms a virtual fist.

- **Thumb Buttons (controllers):**
    - **X (left):** Open quick access toolkit or inventory.
    - **Y (left):** Open game settings menu.
    - **A (right):** Use item in the VR environment; change placement mode.
    - **B (right):** Toggle quick menu.

- **Joysticks (controllers):**
    - **Left joystick:** Move within the VR environment; navigation.
    - **Right joystick:** Rotate in different directions.

`</vr_device_guide>`

Criteria:

`<criteria/>` 1. You can assume that the tools and materials you need are already in your inventory. 2. If you use a trigger, grip, or controller, you must explicitly specify whether it is left, right, or both. 3. If you use a thumb button, you must explicitly state which one (A, B, X, or Y). `</criteria>`

# E   DETAILED EXPERIMENT RESULTS

This section provides comprehensive analysis of our experimental results, including detailed performance breakdowns across models, games, and experimental conditions. We present both aggregated metrics and fine-grained analyses that illuminate specific strengths and weaknesses in current LLMs' ability to reason about VR device manipulations.

## E.1   OVERALL PERFORMANCE ANALYSIS

The table 5 below presents a holistic view of model performance across all experimental conditions. The results reveal a clear performance hierarchy, with Gemini-1.5-Pro achieving the highest average Normalized Step Alignment Score (NSAS) of 0.845, followed closely by GPT-4o (0.832) and GPT-4 (0.824). Notably, even the best-performing models achieve relatively modest Strict Step-by-Step Matching (SSM) scores, with Gemini-1.5-Pro reaching only 8.7% exact sequence matches. This discrepancy between NSAS and SSM scores indicates that while models can identify appropriate actions, they struggle with precise sequencing and complete reproduction of manipulation sequences.

The Sequential Order Preservation (SOP) scores reveal perhaps the most significant challenge facing current LLMs. Even top-performing models achieve SOP scores below 0.3, indicating difficulty in maintaining correct procedural ordering of steps. This limitation is particularly pronounced in zero-shot settings, where SOP scores approach zero for most models, suggesting that procedural reasoning for VR interactions requires exposure to examples rather than emerging from general language understanding.

Human performance provides an important baseline for contextualizing model achievements. While humans achieve comparable NSAS scores (0.817) to top LLMs, they show notably lower SOP scores (0.124) than leading models. This counterintuitive result reflects the challenging nature of the tasks even for experienced VR users and suggests that perfect procedural recall may be less important than adaptive problem-solving in real-world VR interaction.

Table 5: Performance of LLMs across VR Games (Best Few-Shot Setting)

| Model | NSAS | SOP | SSC | SSM | Best FS |
|---|---|---|---|---|---|
| Gemini-1.5-Pro | 0.845 | 0.251 | 0.151 | 0.087 | 5 |
| GPT-4o | 0.832 | 0.291 | 0.190 | 0.135 | 5 |
| GPT-4 | 0.824 | 0.218 | 0.177 | 0.095 | 5 |
| LLaMA-3-8B | 0.823 | 0.283 | 0.200 | 0.040 | 5 |
| Human | 0.817 | 0.124 | 0.174 | 0.021 | - |
| Mixtral-8x7B | 0.790 | 0.123 | 0.142 | 0.039 | 5 |
| GPT-3.5 | 0.778 | 0.169 | 0.137 | 0.037 | 5 |
| GLM-4-Flash | 0.749 | 0.096 | 0.165 | 0.000 | 5 |

## E.2   GAME-SPECIFIC PERFORMANCE PATTERNS

The table 6 below reveals substantial variations in model performance across different VR games, highlighting how game design and interaction complexity influence LLM reasoning capabilities. Vivecraft consistently yields the highest performance across all models, with NSAS scores ranging from 0.909 to 0.938. This strong performance likely reflects the game's discrete, block-based interaction paradigm inherited from Minecraft, which provides clear action-object mappings that align well with linguistic descriptions.

In contrast, Into the Radius proves most challenging, with NSAS scores dropping to 0.618-0.698 across models. This game's emphasis on realistic physics simulation, complex inventory management, and weapon manipulation requires understanding of nuanced spatial relationships and multi-step procedures that current LLMs struggle to capture. The high standard deviation in performance (0.135 for GLM-4-flash) indicates inconsistent model behavior when confronting complex interaction scenarios.

Half-Life: Alyx and Moss: Book II occupy intermediate positions in the difficulty spectrum. Half-Life: Alyx's physics-based puzzles and combat scenarios require precise timing and spatial reasoning,

reflected in extremely low SOP scores (0.022 for GPT-4o). Moss: Book II's third-person perspective and puzzle-platforming elements introduce unique challenges in translating camera-relative directions into controller movements, though models show more consistent performance than in Half-Life: Alyx.

Table 6: Performance comparison across different VR games (5-shot setting). We report NSAS scores (primary metric) and SOP scores (in parentheses).

| Model | Half-Life: Alyx | Radius | Moss | Vivecraft |
|---|---|---|---|---|
| GPT-3.5-turbo | 0.858 (0.123) | 0.662 (0.169) | 0.782 (0.169) | 0.922 (0.043) |
| GPT-4-turbo | 0.852 (0.125) | 0.693 (0.189) | 0.824 (0.218) | 0.927 (0.137) |
| GPT-4o | 0.804 (0.022) | 0.698 (0.291) | 0.824 (0.300) | 0.931 (0.190) |
| Gemini-1.5-Pro | 0.863 (0.209) | 0.682 (0.102) | 0.848 (0.265) | 0.938 (0.250) |
| Mixtral-8x7B | 0.839 (0.126) | 0.666 (0.123) | 0.756 (0.117) | 0.926 (0.060) |
| LLaMA-3-8B | 0.848 (0.126) | 0.644 (0.242) | 0.823 (0.283) | 0.929 (0.039) |
| GLM-4-flash | 0.836 (0.076) | 0.618 (0.096) | 0.749 (0.087) | 0.909 (0.000) |
| Human | 0.845 (0.090) | 0.684 (0.148) | 0.817 (0.112) | 0.935 (0.122) |

### E.3 Impact of Few-Shot Learning

The table 7 below demonstrates the transformative effect of few-shot examples on model performance. The most dramatic improvements occur in SOP scores, which increase by factors of 10-20x from zero-shot to 5-shot settings. GPT-3.5-turbo exemplifies this pattern, improving from 0.036 to 0.226 in SOP F1 score, representing a 527.8% relative gain. This massive improvement suggests that examples primarily help models understand the expected format and level of detail for procedural instructions rather than teaching fundamental VR interaction principles.

The diminishing returns pattern is consistent across models, with the largest gains occurring between zero-shot and 1-shot conditions. The jump from 3-shot to 5-shot provides minimal additional benefit, indicating that models quickly extract relevant patterns from limited examples. Gemini-1.5-Pro shows the most efficient few-shot learning, achieving top performance with fewer examples than competing models, suggesting superior in-context learning capabilities for procedural tasks.

Interestingly, few-shot examples have differential effects across game types. Complex games like Into the Radius show continued improvement with additional examples, while simpler environments like Vivecraft plateau quickly. This pattern indicates that few-shot learning is most beneficial when dealing with diverse interaction patterns and complex procedural sequences.

Table 7: Performance of LLMs across VR Games (Best Few-Shot Setting)

| Model | NSAS | SOP | SSC | SSM | Best FS |
|---|---|---|---|---|---|
| Gemini-1.5-Pro | 0.845 | 0.251 | 0.151 | 0.087 | 5 |
| GPT-4o | 0.832 | 0.291 | 0.190 | 0.135 | 5 |
| GPT-4 | 0.824 | 0.218 | 0.177 | 0.095 | 5 |
| LLaMA-3-8B | 0.823 | 0.283 | 0.200 | 0.040 | 5 |
| Mixtral-8x7B | 0.790 | 0.123 | 0.142 | 0.039 | 5 |
| GPT-3.5 | 0.778 | 0.169 | 0.137 | 0.037 | 5 |
| GLM-4-Flash | 0.749 | 0.096 | 0.165 | 0.000 | 5 |
| Gemini-3-Pro | 0.923 | 0.241 | 0.607 | 0.084 | 3 |
| LLaMA-3-70B | 0.917 | 0.229 | 0.556 | 0.051 | 3 |
| Grok-4 | 0.914 | 0.214 | 0.552 | 0.070 | 3 |
| GPT-5.1 | 0.878 | 0.130 | 0.322 | 0.004 | 5 |
| Human | 0.817 | 0.124 | 0.174 | 0.021 | - |

### E.4 Cognitive Capability Analysis

The figure 3 shows model performance across six cognitive dimensions, revealing distinct capability profiles. All models demonstrate strong task decomposition abilities (7.8-8.5), indicating that breaking down high-level goals into subtasks aligns well with LLMs' training on hierarchical text

structures. Gemini-1.5-Pro leads in this dimension with a score of 8.5, though even smaller models like Mixtral-8x7B achieve respectable scores of 8.0.

Motor action mapping emerges as the most challenging capability across all models (0.5-4.5), highlighting the difficulty of translating abstract action concepts into specific button presses and controller movements. This limitation likely stems from the absence of embodied experience in text-based training data. GPT-4o performs best in this dimension but still falls far short of human-level capability, suggesting a fundamental gap in current architectures.

Procedural reasoning shows high variance across models (2.3-7.0), with Gemini-1.5-Pro again leading. The correlation between procedural reasoning scores and few-shot learning gains suggests that this capability can be partially addressed through examples, though the ceiling remains well below human performance. Spatial reasoning capabilities (4.8-7.5) reveal another significant gap, particularly evident in games requiring 3D navigation and object manipulation.

E.5    STATISTICAL SIGNIFICANCE AND VARIANCE ANALYSIS

The tables 9, 10, 11, 12, 13, 14, 15, 16, 17, 18, 19, and 20 below provide detailed statistical analyses of model performance, revealing important patterns in consistency and reliability. And the figures 7, 8, 9 The standard deviation measurements across different games and shot settings illuminate which models maintain stable performance versus those exhibiting high variability. For instance, in Vivecraft, GPT-3.5-turbo shows remarkably consistent NSAS scores in zero-shot settings (std = 0.0248), but this consistency deteriorates with few-shot examples (std = 0.0734 at 3-shot), suggesting that additional examples introduce uncertainty in the model's approach to task completion.

The variance patterns differ significantly between metrics. NSAS scores generally show lower standard deviations (0.02-0.21 range) compared to SOP scores (0.00-0.34 range), indicating that models more consistently identify relevant steps than maintain proper ordering. This pattern is particularly pronounced in complex games like Into the Radius, where SOP standard deviations exceed 0.3 for several models in few-shot settings. Such high variance suggests that models employ different strategies across different runs, sometimes achieving correct ordering by chance rather than through systematic understanding.

Comparison with human variance provides crucial context for interpreting model stability. Human annotators show standard deviations comparable to mid-tier models (0.084 in cross-game performance), suggesting that some degree of variance is inherent to the task rather than a model limitation. However, humans maintain more consistent SOP performance (std = 0.029) compared to all models except Mixtral-8x7B, indicating more reliable procedural reasoning despite overall lower scores.

Table 8: Average and standard deviation of Normalized Step Alignment Score (NSAS) scores comparison of LLMs on *Vivecraft* under different shot settings.

| Model | GPT-3.5-turbo | | GPT-4-turbo | | GPT-4o | | Gemini-1.5-Pro | | Mixtral-8x7B | | LLaMA-3-8b | | LLaMA-3-70B | | Grok-4 | | GPT-5.1 | | Gemini-3-Pro | |
|---|---|---|---|---|---|---|---|---|---|---|---|---|---|---|---|---|---|---|---|---|
| Metrics | avg | std | avg | std | avg | std | avg | std | avg | std | avg | std | avg | std | avg | std | avg | std | avg | std |
| Zero-shot | 0.9258 | 0.0248 | 0.9255 | 0.0238 | 0.9191 | 0.0306 | 0.9209 | 0.0334 | 0.9312 | 0.0207 | 0.9244 | 0.0329 | 0.9001 | 0.0248 | 0.8956 | 0.0324 | 0.8665 | 0.0645 | 0.8979 | 0.0307 |
| 1-shot | 0.921 | 0.0309 | 0.9349 | 0.0506 | 0.9358 | 0.0735 | 0.9362 | 0.0553 | 0.9219 | 0.0636 | 0.9101 | 0.0765 | 0.8948 | 0.0512 | 0.8680 | 0.1266 | 0.8509 | 0.1077 | 0.8921 | 0.0900 |
| 3-shot | 0.9284 | 0.0734 | 0.914 | 0.1167 | 0.9212 | 0.1115 | 0.9381 | 0.0781 | 0.9005 | 0.1125 | 0.9022 | 0.1051 | 0.9113 | 0.0886 | 0.9217 | 0.1115 | 0.8670 | 0.1358 | 0.9219 | 0.1093 |
| 5-shot | 0.9218 | 0.0385 | 0.9274 | 0.0674 | 0.9305 | 0.0689 | 0.9378 | 0.0708 | 0.9256 | 0.0477 | 0.9289 | 0.0364 | 0.8973 | 0.0587 | 0.8901 | 0.0951 | 0.8638 | 0.0895 | 0.8955 | 0.0742 |

Table 9: Average and standard deviation of Normalized Step Alignment Score (NSAS) scores comparison of LLMs on *Vivecraft* under different shot settings.

| Model | GPT-3.5-turbo | | GPT-4-turbo | | GPT-4o | | Gemini-1.5-Pro | | Mixtral-8x7B | | LLaMA-3-8b | | LLaMA-3-70B | | Grok-4 | | GPT-5.1 | | Gemini-3-Pro | |
|---|---|---|---|---|---|---|---|---|---|---|---|---|---|---|---|---|---|---|---|---|
| Metrics | avg | std | avg | std | avg | std | avg | std | avg | std | avg | std | avg | std | avg | std | avg | std | avg | std |
| Zero-Shot | 0.9258 | 0.0248 | 0.9255 | 0.0238 | 0.9191 | 0.0306 | 0.9209 | 0.0334 | 0.9312 | 0.0207 | 0.9244 | 0.0329 | 0.9001 | 0.0248 | 0.8956 | 0.0324 | 0.8665 | 0.0645 | 0.8979 | 0.0307 |
| 1-shot | 0.921 | 0.0309 | 0.9349 | 0.0506 | 0.9358 | 0.0735 | 0.9362 | 0.0553 | 0.9219 | 0.0636 | 0.9101 | 0.0765 | 0.8948 | 0.0512 | 0.8680 | 0.1266 | 0.8509 | 0.1077 | 0.8921 | 0.0900 |
| 3-shot | 0.9284 | 0.0734 | 0.914 | 0.1167 | 0.9212 | 0.1115 | 0.9381 | 0.0781 | 0.9005 | 0.1125 | 0.9022 | 0.1051 | 0.9113 | 0.0886 | 0.9217 | 0.1115 | 0.8670 | 0.1358 | 0.9219 | 0.1093 |
| 5-shot | 0.9218 | 0.0385 | 0.9274 | 0.0674 | 0.9305 | 0.0689 | 0.9378 | 0.0708 | 0.9256 | 0.0477 | 0.9289 | 0.0364 | 0.8973 | 0.0587 | 0.8901 | 0.0951 | 0.8638 | 0.0895 | 0.8955 | 0.0742 |

Table 10: Average and standard deviation of Sequential Order Preservation (SOP) scores comparison of LLMs on *Vivecraft* under different shot settings.

| Model | GPT-3.5-turbo | | GPT-4-turbo | | GPT-4o | | Gemini-1.5-Pro | | Mixtral-8x7B | | LLaMA-3-8b | | LLaMA-3-70B | | Grok-4 | | GPT-5.1 | | Gemini-3-Pro | |
|---|---|---|---|---|---|---|---|---|---|---|---|---|---|---|---|---|---|---|---|---|
| Metrics | avg | std | avg | std | avg | std | avg | std | avg | std | avg | std | avg | std | avg | std | avg | std | avg | std |
| Zero-Shot | 0.0029 | 0.0312 | 0.0007 | 0.0078 | 0.0012 | 0.0125 | 0.0 | 0.0 | 0.0 | 0.0 | 0.0059 | 0.0624 | 0.0000 | 0.0000 | 0.0022 | 0.0234 | 0.0037 | 0.0280 | 0.0029 | 0.0312 |
| 1-shot | 0.015 | 0.0734 | 0.1203 | 0.2157 | 0.1568 | 0.2337 | 0.1794 | 0.291 | 0.0812 | 0.164 | 0.0351 | 0.1215 | 0.0000 | 0.0000 | 0.0000 | 0.0000 | 0.0037 | 0.0280 | 0.0088 | 0.0569 |
| 3-shot | 0.1302 | 0.2352 | 0.1143 | 0.2136 | 0.2826 | 0.3417 | 0.2335 | 0.3417 | 0.0986 | 0.2024 | 0.1124 | 0.2026 | 0.2817 | 0.3333 | 0.2697 | 0.3192 | 0.1090 | 0.2205 | 0.3790 | 0.3432 |
| 5-shot | 0.0395 | 0.1226 | 0.1366 | 0.2388 | 0.1837 | 0.278 | 0.2495 | 0.3358 | 0.0553 | 0.158 | 0.0374 | 0.1371 | 0.0088 | 0.0937 | 0.0000 | 0.0000 | 0.0000 | 0.0000 | 0.0029 | 0.0312 |

Table 11: Average and standard deviation of Semantic Step Coverage (SSC) scores comparison of LLMs on *Vivecraft* under different shot settings.

| Model | GPT-3.5-turbo | | GPT-4-turbo | | GPT-4o | | Gemini-1.5-Pro | | Mixtral-8x7B | | LLaMA-3-8b | | LLaMA-3-70B | | Grok-4 | | GPT-5.1 | | Gemini-3-Pro | |
|---|---|---|---|---|---|---|---|---|---|---|---|---|---|---|---|---|---|---|---|---|
| Metrics | avg | std | avg | std | avg | std | avg | std | avg | std | avg | std | avg | std | avg | std | avg | std | avg | std |
| Zero-Shot | 0.049 | 0.11 | 0.1301 | 0.1443 | 0.1272 | 0.1511 | 0.2221 | 0.2151 | 0.024 | 0.0999 | 0.1088 | 0.1549 | 0.0661 | 0.1381 | 0.0631 | 0.1283 | 0.0162 | 0.0391 | 0.0900 | 0.1445 |
| 1-shot | 0.1274 | 0.1988 | 0.544 | 0.3672 | 0.6598 | 0.3322 | 0.5747 | 0.3359 | 0.4914 | 0.3617 | 0.3165 | 0.3415 | 0.3078 | 0.1823 | 0.2745 | 0.1944 | 0.0453 | 0.0626 | 0.3293 | 0.1840 |
| 3-shot | 0.4755 | 0.3526 | 0.6486 | 0.3373 | 0.6817 | 0.3204 | 0.6538 | 0.3373 | 0.5414 | 0.37 | 0.5299 | 0.3785 | 0.6174 | 0.3279 | 0.5777 | 0.4096 | 0.4515 | 0.2967 | 0.7147 | 0.3225 |
| 5-shot | 0.18 | 0.2337 | 0.5035 | 0.3772 | 0.6183 | 0.3416 | 0.608 | 0.3546 | 0.3579 | 0.3555 | 0.1606 | 0.2605 | 0.3316 | 0.1931 | 0.3108 | 0.2078 | 0.1437 | 0.1067 | 0.2284 | 0.2213 |

Table 12: Average and standard deviation of Normalized Step Alignment Score (NSAS) scores comparison of LLMs on *Half-Life: Alyx* under different shot settings.

| Model | GPT-3.5-turbo | | GPT-4-turbo | | GPT-4o | | Gemini-1.5-Pro | | Mixtral-8x7B | | LLaMA-3-8b | | LLaMA-3-70B | | Grok-4 | | GPT-5.1 | | Gemini-3-Pro | |
|---|---|---|---|---|---|---|---|---|---|---|---|---|---|---|---|---|---|---|---|---|
| Metrics | avg | std | avg | std | avg | std | avg | std | avg | std | avg | std | avg | std | avg | std | avg | std | avg | std |
| Zero-Shot | 0.838 | 0.0413 | 0.8456 | 0.0366 | 0.8376 | 0.0424 | 0.8447 | 0.032 | 0.8376 | 0.0331 | 0.848 | 0.0317 | 0.9039 | 0.0203 | 0.9095 | 0.0198 | 0.8523 | 0.0550 | 0.9136 | 0.0200 |
| 1-shot | 0.8354 | 0.0582 | 0.8427 | 0.0489 | 0.8472 | 0.0629 | 0.8627 | 0.0482 | 0.807 | 0.1099 | 0.8131 | 0.1289 | 0.9077 | 0.0238 | 0.9147 | 0.0271 | 0.8724 | 0.0691 | 0.9231 | 0.0306 |
| 3-shot | 0.8452 | 0.0551 | 0.845 | 0.0467 | 0.838 | 0.0757 | 0.8701 | 0.0603 | 0.8255 | 0.0819 | 0.8449 | 0.0707 | 0.9191 | 0.0263 | 0.9203 | 0.0301 | 0.8844 | 0.0673 | 0.9260 | 0.0321 |
| 5-shot | 0.8577 | 0.0773 | 0.8523 | 0.0613 | 0.8039 | 0.0694 | 0.8625 | 0.0691 | 0.8394 | 0.0834 | 0.848 | 0.0976 | 0.9278 | 0.0308 | 0.9245 | 0.0392 | 0.9030 | 0.0508 | 0.9292 | 0.0384 |

Table 13: Average and standard deviation of Sequential Order Preservation (SOP) scores comparison of LLMs on *Half-Life: Alyx* under different shot settings.

| Model | GPT-3.5-turbo | | GPT-4-turbo | | GPT-4o | | Gemini-1.5-Pro | | Mixtral-8x7B | | LLaMA-3-8b | | LLaMA-3-70B | | Grok-4 | | GPT-5.1 | | Gemini-3-Pro | |
|---|---|---|---|---|---|---|---|---|---|---|---|---|---|---|---|---|---|---|---|---|
| Metrics | avg | std | avg | std | avg | std | avg | std | avg | std | avg | std | avg | std | avg | std | avg | std | avg | std |
| Zero-Shot | 0.0098 | 0.0802 | 0.0252 | 0.1265 | 0.0396 | 0.1745 | 0.0082 | 0.0669 | 0.0019 | 0.0158 | 0.0123 | 0.0704 | 0.0000 | 0.0000 | 0.0226 | 0.1277 | 0.0067 | 0.0316 | 0.0334 | 0.1698 |
| 1-shot | 0.0447 | 0.0764 | 0.0402 | 0.1224 | 0.024 | 0.0733 | 0.0198 | 0.1263 | 0.0425 | 0.0816 | 0.0447 | 0.0967 | 0.0633 | 0.0896 | 0.1418 | 0.2141 | 0.0562 | 0.1164 | 0.1370 | 0.1764 |
| 3-shot | 0.0725 | 0.1159 | 0.0312 | 0.0725 | 0.0701 | 0.1261 | 0.1349 | 0.2187 | 0.0703 | 0.1094 | 0.087 | 0.1687 | 0.1523 | 0.2214 | 0.1801 | 0.2663 | 0.0975 | 0.1335 | 0.1478 | 0.2045 |
| 5-shot | 0.123 | 0.1834 | 0.1248 | 0.2382 | 0.0216 | 0.0809 | 0.2089 | 0.2938 | 0.1257 | 0.2409 | 0.1259 | 0.2385 | 0.2515 | 0.3235 | 0.3509 | 0.3824 | 0.2511 | 0.3231 | 0.3095 | 0.3807 |

Table 14: Average and standard deviation of Semantic Step Coverage (SSC) scores comparison of LLMs on *Half-Life: Alyx* under different shot settings.

| Model | GPT-3.5-turbo | | GPT-4-turbo | | GPT-4o | | Gemini-1.5-Pro | | Mixtral-8x7B | | LLaMA-3-8b | | LLaMA-3-70B | | Grok-4 | | GPT-5.1 | | Gemini-3-Pro | |
|---|---|---|---|---|---|---|---|---|---|---|---|---|---|---|---|---|---|---|---|---|
| Metrics | avg | std | avg | std | avg | std | avg | std | avg | std | avg | std | avg | std | avg | std | avg | std | avg | std |
| Zero-Shot | 0.0785 | 0.1843 | 0.2231 | 0.2089 | 0.2424 | 0.2111 | 0.1989 | 0.1982 | 0.0716 | 0.1187 | 0.1662 | 0.172 | 0.2676 | 0.2473 | 0.3874 | 0.2736 | 0.1693 | 0.1449 | 0.4934 | 0.2915 |
| 1-shot | 0.2562 | 0.235 | 0.3485 | 0.2336 | 0.4184 | 0.2413 | 0.3859 | 0.2654 | 0.3256 | 0.1934 | 0.3872 | 0.2058 | 0.4790 | 0.2759 | 0.5427 | 0.2460 | 0.3417 | 0.2254 | 0.5525 | 0.3053 |
| 3-shot | 0.3072 | 0.2444 | 0.3648 | 0.2414 | 0.5611 | 0.229 | 0.5494 | 0.2887 | 0.3544 | 0.2202 | 0.4599 | 0.2371 | 0.6075 | 0.2956 | 0.5898 | 0.2581 | 0.3919 | 0.2003 | 0.5798 | 0.3317 |
| 5-shot | 0.425 | 0.2814 | 0.6127 | 0.2856 | 0.6934 | 0.2359 | 0.6299 | 0.315 | 0.4642 | 0.2957 | 0.5152 | 0.2708 | 0.6920 | 0.2518 | 0.6555 | 0.2914 | 0.4929 | 0.2721 | 0.6502 | 0.3073 |

Table 15: Normalized Step Alignment Score (NSAS) scores comparison of LLMs on *Moss: Book II* under different shot settings

| Model | GPT-3.5-turbo | | GPT-4-turbo | | GPT-4o | | Gemini-1.5-Pro | | Mixtral-8x7B | | LLaMA-3-8b | | LLaMA-3-70B | | Grok-4 | | GPT-5.1 | | Gemini-3-Pro | |
|---|---|---|---|---|---|---|---|---|---|---|---|---|---|---|---|---|---|---|---|---|
| Metrics | avg | std | avg | std | avg | std | avg | std | avg | std | avg | std | avg | std | avg | std | avg | std | avg | std |
| Zero-Shot | 0.7819 | 0.0403 | 0.8055 | 0.0717 | 0.7871 | 0.0596 | 0.7994 | 0.0548 | 0.7913 | 0.0595 | 0.7916 | 0.0572 | 0.8916 | 0.0224 | 0.8875 | 0.0269 | 0.8074 | 0.0811 | 0.8965 | 0.0255 |
| 1-shot | 0.776 | 0.0616 | 0.7993 | 0.0771 | 0.803 | 0.0924 | 0.8139 | 0.0778 | 0.7663 | 0.0793 | 0.7938 | 0.0763 | 0.9075 | 0.0267 | 0.8968 | 0.0416 | 0.8393 | 0.0880 | 0.9151 | 0.0290 |
| 3-shot | 0.7776 | 0.0889 | 0.818 | 0.0925 | 0.8016 | 0.1242 | 0.8302 | 0.0935 | 0.7613 | 0.1341 | 0.7895 | 0.1371 | 0.9199 | 0.0392 | 0.9122 | 0.0438 | 0.8773 | 0.0559 | 0.9237 | 0.0339 |
| 5-shot | 0.782 | 0.0952 | 0.8243 | 0.102 | 0.8237 | 0.1092 | 0.8478 | 0.1017 | 0.756 | 0.1469 | 0.8232 | 0.105 | 0.9241 | 0.0406 | 0.9178 | 0.0417 | 0.8883 | 0.0664 | 0.9282 | 0.0354 |

Table 16: Average and standard deviation of Sequential Order Preservation (SOP) scores comparison of LLMs on *Moss: Book II* under different shot settings.

| Model | GPT-3.5-turbo | | GPT-4-turbo | | GPT-4o | | Gemini-1.5-Pro | | Mixtral-8x7B | | LLaMA-3-8b | | LLaMA-3-70B | | Grok-4 | | GPT-5.1 | | Gemini-3-Pro | |
|---|---|---|---|---|---|---|---|---|---|---|---|---|---|---|---|---|---|---|---|---|
| Metrics | avg | std | avg | std | avg | std | avg | std | avg | std | avg | std | avg | std | avg | std | avg | std | avg | std |
| Zero-Shot | 0.0091 | 0.0581 | 0.0263 | 0.1533 | 0.0113 | 0.0494 | 0.0197 | 0.1029 | 0.0052 | 0.033 | 0.0 | 0.0 | 0.0000 | 0.0000 | 0.0000 | 0.0000 | 0.0000 | 0.0000 | 0.0119 | 0.0762 |
| 1-shot | 0.034 | 0.1568 | 0.0663 | 0.2044 | 0.1084 | 0.2486 | 0.1145 | 0.2495 | 0.0739 | 0.1721 | 0.0578 | 0.1486 | 0.0759 | 0.1942 | 0.1062 | 0.2374 | 0.0456 | 0.1038 | 0.0887 | 0.2098 |
| 3-shot | 0.1581 | 0.242 | 0.1678 | 0.2522 | 0.2324 | 0.3185 | 0.2272 | 0.3324 | 0.1351 | 0.252 | 0.2584 | 0.3089 | 0.2759 | 0.3045 | 0.2060 | 0.2904 | 0.1481 | 0.2026 | 0.2298 | 0.3019 |
| 5-shot | 0.1686 | 0.244 | 0.2182 | 0.2801 | 0.2998 | 0.3062 | 0.2652 | 0.3596 | 0.1169 | 0.247 | 0.2831 | 0.3097 | 0.2703 | 0.3077 | 0.2310 | 0.3010 | 0.2056 | 0.2822 | 0.2616 | 0.3151 |

Table 17: Average and standard deviation of Semantic Step Coverage (SSC) scores comparison of LLMs on *Moss: Book II* under different shot settings.

| Model | GPT-3.5-turbo | | GPT-4-turbo | | GPT-4o | | Gemini-1.5-Pro | | Mixtral-8x7B | | LLaMA-3-8b | | LLaMA-3-70B | | Grok-4 | | GPT-5.1 | | Gemini-3-Pro | |
|---|---|---|---|---|---|---|---|---|---|---|---|---|---|---|---|---|---|---|---|---|
| Metrics | avg | std | avg | std | avg | std | avg | std | avg | std | avg | std | avg | std | avg | std | avg | std | avg | std |
| Zero-Shot | 0.0715 | 0.1792 | 0.2491 | 0.2844 | 0.2313 | 0.2567 | 0.1763 | 0.221 | 0.0407 | 0.0991 | 0.1208 | 0.1779 | 0.1401 | 0.1876 | 0.1286 | 0.1794 | 0.0205 | 0.0470 | 0.2099 | 0.2192 |
| 1-shot | 0.0748 | 0.1719 | 0.259 | 0.2771 | 0.3682 | 0.3018 | 0.3449 | 0.3396 | 0.1749 | 0.2393 | 0.2319 | 0.293 | 0.3421 | 0.2902 | 0.3319 | 0.2375 | 0.1857 | 0.1592 | 0.3717 | 0.2929 |
| 3-shot | 0.3349 | 0.3069 | 0.4593 | 0.3309 | 0.5001 | 0.3444 | 0.5238 | 0.3738 | 0.3207 | 0.3105 | 0.4689 | 0.3399 | 0.4884 | 0.3352 | 0.5130 | 0.2916 | 0.2986 | 0.2235 | 0.5575 | 0.3392 |
| 5-shot | 0.3737 | 0.3213 | 0.4951 | 0.3373 | 0.5562 | 0.3319 | 0.6091 | 0.3476 | 0.2974 | 0.3385 | 0.4567 | 0.3173 | 0.5416 | 0.3313 | 0.5583 | 0.3393 | 0.3834 | 0.2477 | 0.5715 | 0.3281 |

Table 18: Average and standard deviation of Normalized Step Alignment Score (NSAS) scores comparison of LLMs on *Into the Radius* under different shot settings.

| Model | GPT-3.5-turbo | | GPT-4-turbo | | GPT-4o | | Gemini-1.5-Pro | | Mixtral-8x7B | | LLaMA-3-8b | | LLaMA-3-70B | | Grok-4 | | GPT-5.1 | | Gemini-3-Pro | |
|---|---|---|---|---|---|---|---|---|---|---|---|---|---|---|---|---|---|---|---|---|
| Metrics | avg | std | avg | std | avg | std | avg | std | avg | std | avg | std | avg | std | avg | std | avg | std | avg | std |
| Zero-Shot | 0.6165 | 0.0755 | 0.6408 | 0.0955 | 0.5939 | 0.1306 | 0.6492 | 0.1018 | 0.6644 | 0.0718 | 0.6447 | 0.0882 | 0.8975 | 0.0226 | 0.8891 | 0.0411 | 0.7947 | 0.0774 | 0.9085 | 0.0337 |
| 1-shot | 0.641 | 0.1177 | 0.6519 | 0.1421 | 0.6282 | 0.1687 | 0.6875 | 0.1159 | 0.6285 | 0.1684 | 0.6285 | 0.1346 | 0.9070 | 0.0345 | 0.8719 | 0.0276 | 0.8315 | 0.0659 | 0.9133 | 0.0357 |
| 3-shot | 0.6305 | 0.128 | 0.6802 | 0.1645 | 0.6491 | 0.2057 | 0.6634 | 0.1638 | 0.618 | 0.1633 | 0.6479 | 0.1606 | 0.9165 | 0.0327 | 0.9017 | 0.0630 | 0.8483 | 0.0599 | 0.9219 | 0.0375 |
| 5-shot | 0.6621 | 0.1291 | 0.6927 | 0.1721 | 0.6984 | 0.2136 | 0.6818 | 0.1191 | 0.666 | 0.1495 | 0.6443 | 0.211 | 0.9165 | 0.0410 | 0.9112 | 0.0484 | 0.8573 | 0.0687 | 0.9265 | 0.0485 |

Table 19: Average and standard deviation of Sequential Order Preservation (SOP) scores comparison of LLMs on *Into the Radius* under different shot settings.

| Model | GPT-3.5-turbo | | GPT-4-turbo | | GPT-4o | | Gemini-1.5-Pro | | Mixtral-8x7B | | LLaMA-3-8b | | LLaMA-3-70B | | Grok-4 | | GPT-5.1 | | Gemini-3-Pro | |
|---|---|---|---|---|---|---|---|---|---|---|---|---|---|---|---|---|---|---|---|---|
| Metrics | avg | std | avg | std | avg | std | avg | std | avg | std | avg | std | avg | std | avg | std | avg | std | avg | std |
| Zero-Shot | 0.0091 | 0.0581 | 0.0263 | 0.1533 | 0.0113 | 0.0494 | 0.0197 | 0.1029 | 0.0052 | 0.033 | 0.0 | 0.0 | 0.0123 | 0.0630 | 0.0370 | 0.1386 | 0.0000 | 0.0000 | 0.0401 | 0.1528 |
| 1-shot | 0.034 | 0.1568 | 0.0663 | 0.2044 | 0.1084 | 0.2486 | 0.1145 | 0.2495 | 0.0739 | 0.1721 | 0.0578 | 0.1486 | 0.1799 | 0.2850 | 0.0000 | 0.0000 | 0.0404 | 0.1387 | 0.1205 | 0.2522 |
| 3-shot | 0.1581 | 0.242 | 0.1678 | 0.2522 | 0.2324 | 0.3185 | 0.2272 | 0.3324 | 0.1351 | 0.252 | 0.2584 | 0.3089 | 0.2053 | 0.2758 | 0.2003 | 0.3009 | 0.0662 | 0.1330 | 0.2081 | 0.2859 |
| 5-shot | 0.1686 | 0.244 | 0.2182 | 0.2801 | 0.2998 | 0.3062 | 0.2652 | 0.3596 | 0.1169 | 0.247 | 0.2831 | 0.3097 | 0.2321 | 0.3149 | 0.3203 | 0.3509 | 0.0622 | 0.1455 | 0.2804 | 0.3294 |

Table 20: Average and standard deviation of Semantic Step Coverage (SSC) scores comparison of LLMs on *Into the Radius* under different shot settings.

| Model | GPT-3.5-turbo | | GPT-4-turbo | | GPT-4o | | Gemini-1.5-Pro | | Mixtral-8x7B | | LLaMA-3-8b | | LLaMA-3-70B | | Grok-4 | | GPT-5.1 | | Gemini-3-Pro | |
|---|---|---|---|---|---|---|---|---|---|---|---|---|---|---|---|---|---|---|---|---|
| Metrics | avg | std | avg | std | avg | std | avg | std | avg | std | avg | std | avg | std | avg | std | avg | std | avg | std |
| Zero-Shot | 0.0354 | 0.0982 | 0.1528 | 0.1774 | 0.2199 | 0.1745 | 0.1783 | 0.2107 | 0.0406 | 0.0899 | 0.1243 | 0.1655 | 0.1671 | 0.2225 | 0.3100 | 0.2305 | 0.0945 | 0.0945 | 0.4250 | 0.2078 |
| 1-shot | 0.1511 | 0.2246 | 0.304 | 0.2983 | 0.4102 | 0.2585 | 0.2823 | 0.3277 | 0.2593 | 0.3249 | 0.3171 | 0.269 | 0.4190 | 0.3079 | 0.4098 | 0.2744 | 0.1565 | 0.1162 | 0.5220 | 0.2739 |
| 3-shot | 0.2321 | 0.2713 | 0.4463 | 0.3099 | 0.5623 | 0.2976 | 0.3402 | 0.3379 | 0.3544 | 0.2621 | 0.5319 | 0.2382 | 0.5113 | 0.2765 | 0.5294 | 0.2839 | 0.2220 | 0.1755 | 0.5766 | 0.3039 |
| 5-shot | 0.3302 | 0.3115 | 0.5082 | 0.3063 | 0.6194 | 0.2877 | 0.2971 | 0.3187 | 0.285 | 0.3384 | 0.5314 | 0.2886 | 0.5601 | 0.3133 | 0.5638 | 0.3041 | 0.2692 | 0.1803 | 0.6109 | 0.3173 |

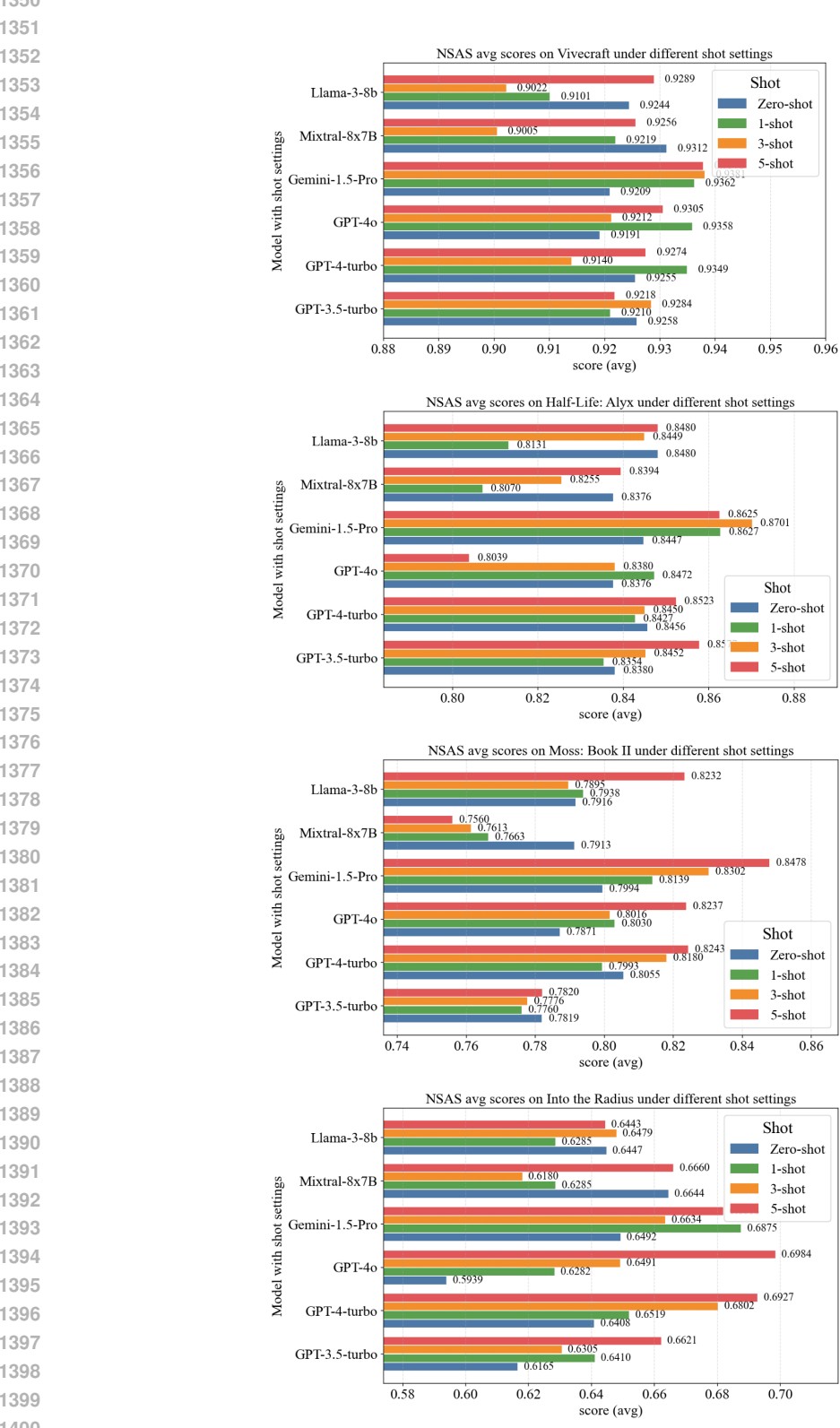

Figure 7: LLMs NSAS (avg) by Different Shot Setting Across Four VR Games

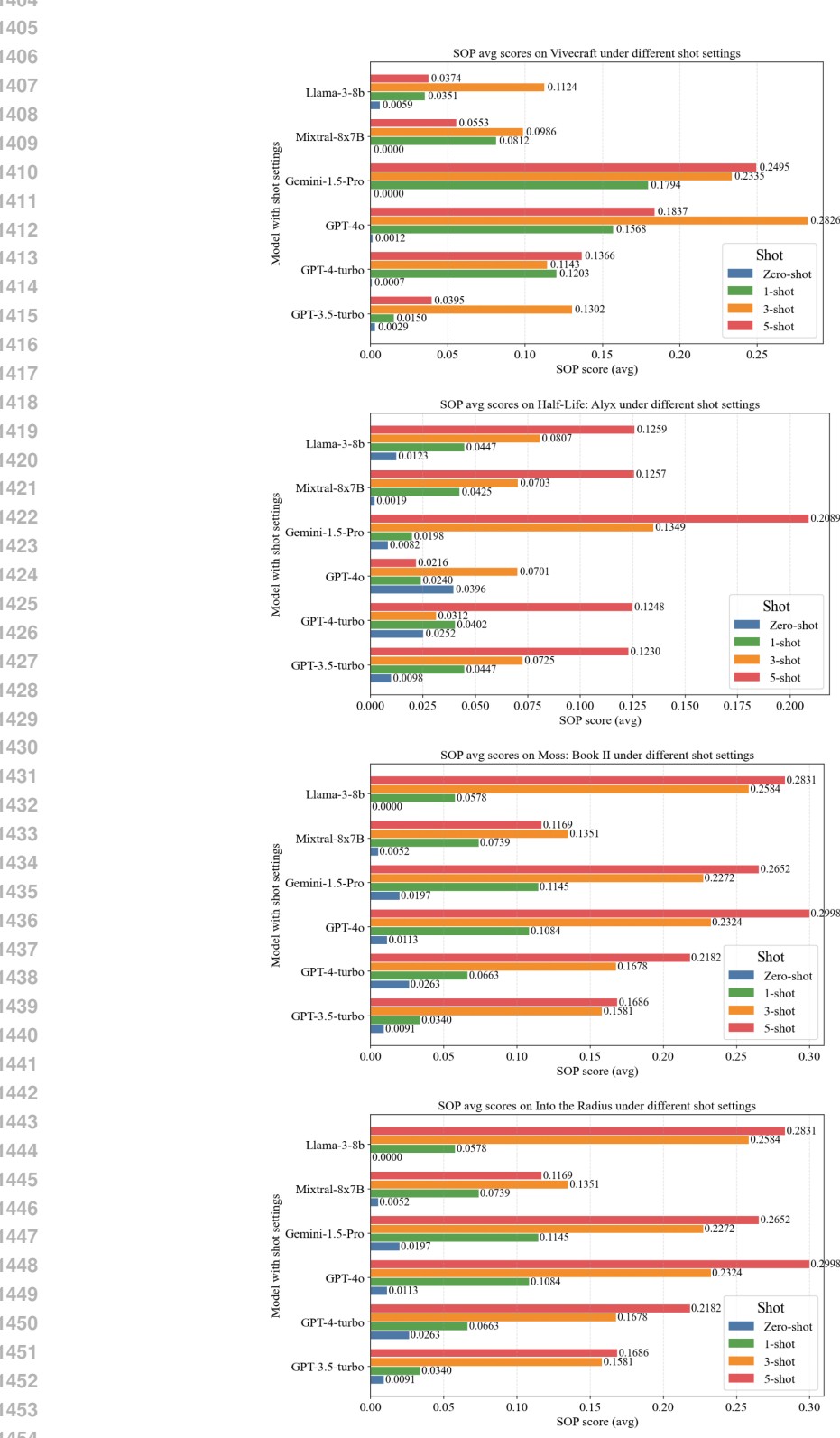

Figure 8: LLMs SOP (avg) by Different Shot Setting Across Four VR Games

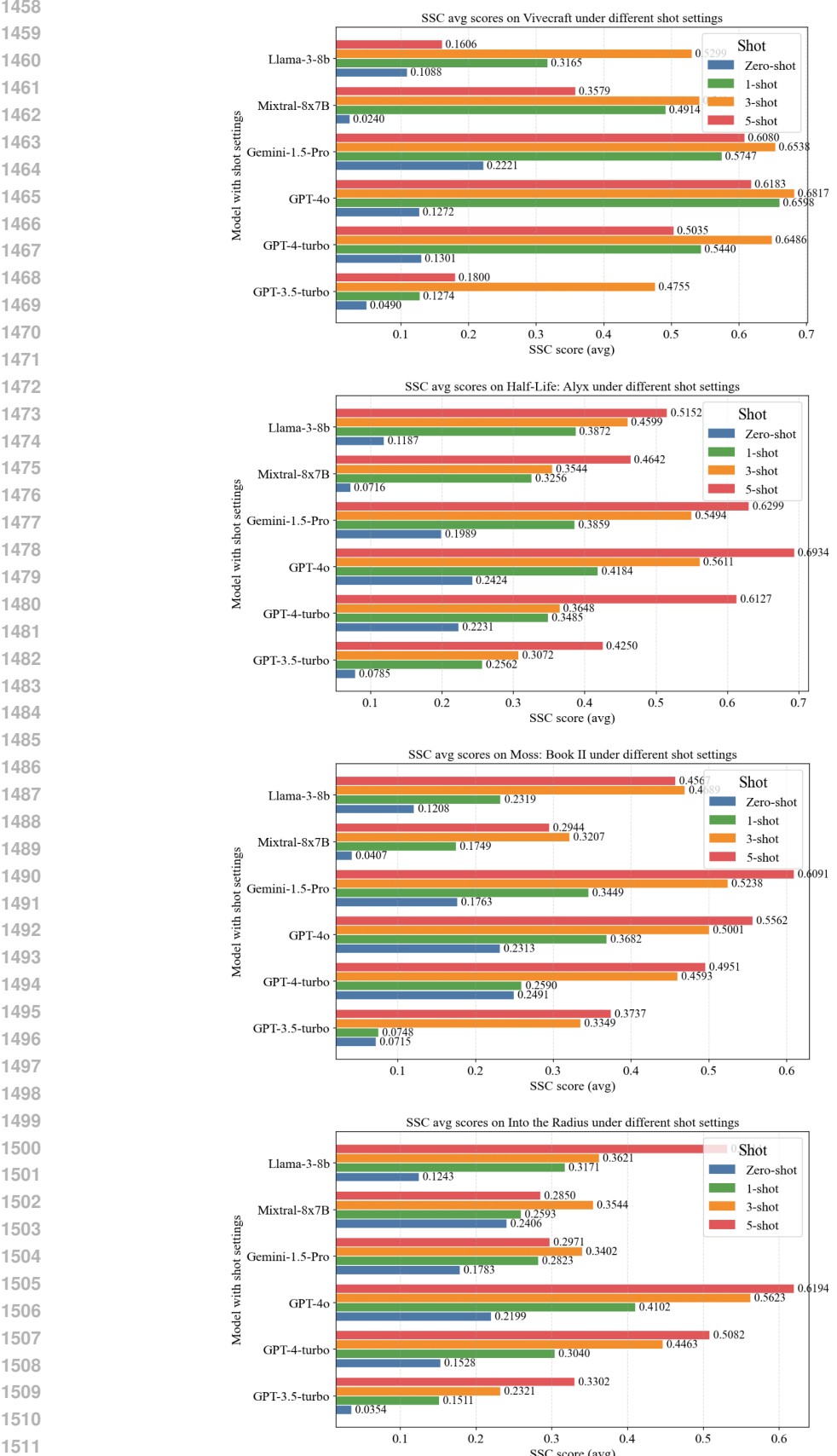

Figure 9: LLMs SSC (avg) by Different Shot Setting Across Four VR Games

### E.6 CROSS-GAME GENERALIZATION PATTERNS

The cross-game performance analysis reveals important insights about model generalization capabilities. Models that perform well on one game do not necessarily maintain their advantage across others. For example, while GPT-4o achieves the highest SOP score in Into the Radius (0.291), it performs poorly in Half-Life: Alyx (0.022). This game-specific variation suggests that models may overfit to particular interaction patterns rather than developing general VR manipulation capabilities.

The "Game Gap" metric in the table 3 quantifies this generalization challenge. Lower values indicate more consistent cross-game performance. Mixtral-8x7B achieves the lowest Game Gap (0.070), despite not leading in any individual game. This consistency might make it more suitable for applications requiring reliable performance across diverse VR experiences. In contrast, GPT-4o's high Game Gap (0.127) reflects its specialized strengths and weaknesses across different interaction paradigms.

Analysis of confusion patterns reveals that models struggle most when transitioning between games with different control schemes. The shift from Vivecraft's discrete block interactions to Half-Life: Alyx's continuous physics manipulation represents a fundamental change in how actions map to controller inputs. Models trained primarily on text lack the embodied experience to navigate these transitions smoothly, often applying inappropriate interaction patterns learned from one context to another.

### E.7 TEMPORAL DYNAMICS IN SEQUENTIAL TASKS

Detailed examination of step-by-step performance reveals how models handle temporal dependencies in VR interactions. Early steps in sequences generally show higher accuracy (NSAS > 0.9) across all models, with performance degrading for later steps. This degradation is particularly severe for steps that depend on the successful completion of previous actions. For instance, in a sequence like "pick up object, aim at target, throw object," models may correctly identify all three actions but fail to recognize that aiming requires successfully completing the pickup action first.

The SOP metric specifically captures these temporal dependencies, and the low scores across all models highlight a fundamental limitation in current architectures. Even with few-shot examples that demonstrate correct ordering, models struggle to internalize the causal relationships between steps. This suggests that improved performance may require architectural innovations that better capture temporal and causal reasoning, rather than simply scaling existing approaches.

Error analysis reveals common patterns in temporal mistakes. Models frequently suggest parallel actions that must be performed sequentially (e.g., "press trigger while reaching for object" when the trigger can only be meaningfully pressed after grasping). They also struggle with iterative processes, often omitting loop conditions or termination criteria. These patterns indicate that models lack an understanding of the physical constraints that govern VR interactions.

### E.8 DETAILED PERFORMANCE TABLES AND VISUALIZATIONS

The table 2 provides granular data for researchers seeking to understand specific model behaviors. These tables reveal several noteworthy patterns. First, the relationship between different metrics is non-linear. High NSAS scores do not guarantee good SOP performance, and models with similar average scores may achieve them through different strengths. This multidimensional performance landscape suggests that selecting models for specific applications requires careful consideration of which capabilities are most critical.

The table 2 illustrates the strict matching process, highlighting why SSM scores remain low even for generally capable models. The requirement for exact sequence length and step-by-step correspondence proves extremely demanding. Even minor variations in phrasing or step granularity result in match failures. This visualization helps explain why SSM may be overly strict for practical applications, where functional equivalence matters more than exact replication.

The table 2 demonstrates the more nuanced evaluation approach that underlies our NSAS and SOP metrics. By identifying the longest common subsequences with semantic matching, these metrics better capture functional understanding while still penalizing significant deviations from ground truth.

The visualization shows how models might achieve reasonable NSAS scores by identifying most relevant actions while still failing SOP evaluation due to ordering errors.

The heat maps of model performance across game-task combinations reveal clustering of difficulty. Certain task types (e.g., combat sequences in Half-Life: Alyx, inventory management in Into the Radius) consistently challenge all models, while others (e.g., block placement in Vivecraft) show near-ceiling performance. These patterns suggest that targeted improvements for specific interaction types might yield better results than general capability enhancement.

### E.9 IMPLICATIONS FOR FUTURE RESEARCH

The detailed experimental results paint a complex picture of current LLM capabilities and limitations in VR interaction reasoning. While models demonstrate competence in identifying relevant actions and decomposing high-level goals, they consistently struggle with the procedural and embodied aspects of VR interaction. The strong effect of few-shot examples suggests that current models possess latent capabilities that can be activated through appropriate prompting, but fundamental architectural limitations prevent them from achieving human-like understanding of physical manipulation sequences.

The high variance in performance across games and tasks indicates that robustness remains a significant challenge. Models that excel in one context may fail dramatically in another, limiting their practical applicability. This brittleness likely stems from the discrete nature of text-based training, which lacks the continuous, embodied experience that humans leverage when learning new physical tasks.

Moving forward, these results suggest several promising research directions. Multimodal models that incorporate visual and proprioceptive information alongside text may better capture the embodied nature of VR interactions. Explicit modeling of temporal and causal relationships could address the procedural reasoning gaps identified in our experiments. Finally, training on synthetic VR interaction data or through simulated embodiment might provide models with the experiential knowledge currently lacking in text-only approaches.

The detailed results also highlight the importance of comprehensive evaluation frameworks that assess multiple dimensions of capability. Single metrics fail to capture the complexity of VR interaction reasoning, and future benchmarks should continue to embrace multidimensional evaluation approaches that can identify specific strengths and weaknesses in model capabilities.

## F DISCUSSION, LIMITATIONS & BROADER IMPACTS

Our investigation into LLMs' ability to translate semantic actions into VR device manipulations reveals both promising capabilities and fundamental limitations that reflect broader challenges in bridging linguistic understanding and embodied interaction. The relatively low Sequential Order Preservation (SOP) scores across all evaluated models indicate that current LLMs struggle with the temporal reasoning required for complex procedural tasks. This limitation suggests that while LLMs can identify relevant actions and understand their purposes, they lack the embodied experience necessary to accurately sequence physical manipulations.

The substantial performance variations across different VR games highlight how interaction complexity and consistency impact model performance. Our primary goal was to first establish a robust and comprehensive benchmark on a diverse set of known games.Games with standardized, discrete actions (like Vivecraft's block-based interactions) prove more amenable to LLM reasoning than those requiring nuanced controller movements or complex spatial reasoning (like Half-Life: Alyx). This pattern suggests that current language models may benefit from more structured representations of physical actions and explicit training on procedural sequences.

The significant improvement from few-shot examples demonstrates that LLMs possess latent capabilities for VR interaction reasoning that can be activated through appropriate prompting. However, the fact that performance plateaus with additional examples indicates fundamental architectural limitations rather than simple lack of exposure to relevant examples. This finding suggests that advances in VR-capable AI may require new training paradigms that incorporate spatial and temporal reasoning more directly.

From a broader perspective, this work carries important implications for the future of human-computer interaction and AI development. On the positive side, LLMs that can effectively reason about VR interactions could dramatically improve accessibility for users with motor impairments, enable more intuitive natural language interfaces for VR applications, and accelerate the development of intelligent tutoring systems for VR training scenarios. The potential transfer of these capabilities to robotic systems could enable more sophisticated human-robot collaboration in both virtual and physical environments.

However, we must also consider potential negative implications. As LLMs gain greater agency in controlling virtual (and potentially physical) systems, questions of safety, security, and user autonomy become paramount. The ability to translate high-level commands into detailed manipulation sequences could be exploited for unauthorized system control or social engineering attacks. Additionally, the computational resources required for training and deploying such models raise environmental concerns that must be balanced against their benefits.

The digital divide may be exacerbated as advanced VR-AI systems require substantial hardware investments and technical expertise. Ensuring equitable access to these technologies will require conscious effort from researchers, developers, and policymakers. Privacy concerns also emerge as these systems necessarily monitor and analyze detailed user movement patterns and interaction behaviors.

Moving forward, the field must pursue responsible development practices that prioritize user safety, privacy, and autonomy while advancing the technical capabilities of VR-AI systems. This includes developing robust evaluation frameworks that assess not only task performance but also failure modes, implementing transparent systems that users can understand and control, and ensuring that advances in VR interaction AI serve to augment rather than replace human agency in virtual environments.

## G    LARGE LANGUAGE MODELS USAGE STATEMENT

This work incorporated LLMs to aid in editorial refinement and linguistic improvement of the manuscript. The models provided assistance with stylistic enhancements and clarity optimization, including tasks such as rephrasing sentences and correcting grammatical errors.

We explicitly note that LLMs played no role in the conceptualization, theoretical development, or experimental design aspects of this research. The authors retain full responsibility for the entirety of the manuscript's content, including sections improved with LLM support. All LLM-assisted text has been carefully reviewed to ensure adherence to academic standards and ethical research practices.

