# OpenReview forum: "ComboBench: Can LLMs Manipulate Physical Devices to Play Virtual Reality Games?"
_ICLR.cc/2026/Conference — Submitted to ICLR 2026_

### Official Review · Reviewer_19Um · 2025-10-31

**Soundness:** 2
**Presentation:** 4
**Contribution:** 2
**Rating:** 4
**Confidence:** 4

**Summary:**

This paper presents ComboBench, a benchmark designed to evaluate an LLM's ability to translate high-level actions into VR manipulations across 262 scenarios. The authors evaluate several state-of-the-art LLMs and report that while they excel at task decomposition, they struggle significantly with procedural reasoning and motor mapping. However, the study's conclusions are undermined by a severely rigid evaluation framework that relies on a single ground-truth sequence for each task. This design leads to extremely low scores even for the human baseline (e.g., 1.2% SSM), suggesting the metrics measure similarity to one specific annotation rather than actual task success, thus calling the validity of the reported model capabilities into question.

**Strengths:**

1. The paper focuses on the crucial transition from "understanding" to "action" for LLMs, specifically how high-level intent is translated into low-level physical device operations. This is a frontier and highly valuable research direction in embodied AI and human-computer interaction, especially within immersive environments like VR.

2. The authors have constructed and committed to releasing the ComboBench dataset. The process for its creation (game selection, expert annotation, LLM-assisted labeling) is clear. This provides a valuable public resource for the community further research.

**Weaknesses:**

1. The paper’s evaluation framework suffers from severe rigidity problem. As a result, even the human baseline attains extremely low scores (e.g., SSM only 1.2%), strongly suggesting that the metric fails to measure task success and instead measures similarity to a particular annotation. Moreover, the ground truth relies on a single action sequence, while VR tasks typically allow multiple valid solutions. This “single truth” design unfairly penalizes models that produce other valid action sequences, distorting the true assessment of model capability.

2. Comparing models with humans is important to the paper’s argument, but the methodology is flawed. Having human participants read instructions and “write down” action steps is a task of memory retrieval and language expression, not an actual, dynamic interaction in a VR environment. These two tasks rely on different cognitive abilities and constraints, which likely explains the counterintuitive result that “models outperform humans on certain procedural metrics.” Therefore, the current human baseline cannot serve as a valid reference for real-world VR interaction ability.

3. The chosen open-source models (Llama-3-8B, Mixtral-8x7B) differ vastly in parameter scale from state-of-the-art closed-source models (GPT-4o, Gemini-1.5-Pro). While this reflects the current model landscape, directly comparing 8B models to hundred-billion–parameter (or larger) models will mingle “scaling effects” with “architectural differences.” Testing models with similar architectures but different sizes (e.g., Llama-3-70B) would make the conclusions more compelling.

**Questions:**

1. On the human baseline: Could you clarify the data collection procedure? If participants indeed wrote down steps, how do you justify this disembodied evaluation as a valid proxy for VR interaction ability, especially given the extremely low human scores under this paradigm?

2. What is the specific method or equations that map base evaluation metrics (NSAS, SOP, etc.) to the six cognitive ability scores in Figure 1?

3. Why did you only experiment with temperature = 0, and did you explore the impact of non-zero temperatures on generating valid action sequences?

4. Given the paper’s focus on cognitive abilities, why was there no evaluation of advanced prompting strategies such as Chain-of-Thought and their potential to improve performance?

5. What is the reason for selecting the specific similarity threshold of 0.8387?

---

### Official Review · Reviewer_gDtt · 2025-10-31

**Soundness:** 2
**Presentation:** 1
**Contribution:** 2
**Rating:** 2
**Confidence:** 4

**Summary:**

This paper proposes a benchmark to evaluate LLMs' ability to play VR games. The benchmark includes 262 scenarios from 4 VR games, and categorizes the actions into different labels for analyzing different aspects of model abilities. Some evaluation metrics are designed for comparing LLM-generated steps versus ground truth. This paper tested 6 LLMs on this benchmark and identified some problems of existing models.

**Strengths:**

The idea of testing LLM in VR is interesting. Labeling each action with semantics is a good way for analyzing model behaviors.

**Weaknesses:**

1. **Motivation.** The motivation of letting LLMs manipulate in VR is unclear. Is this benchmark mainly testing LLMs in long-horizon tasks? But there are many other more meaningful tasks like robotics, digital agents, etc to test such capability. Anything unique in this VR setting?

2. **Evaluation Metrics.** The proposed metrics are mainly compare the difference of action sequences between model and ground-truth. However, for such long-horizon tasks, there could be multiple trajectories lead to the success end state. Therefore, having metrics for task success rate is very important, but missing in this paper. It is hard to interpret the results using such step matching metrics. In addition, some results show models perform better than human (Table 1), which means human may do the task in a different way but still achieve the goal, and therefore those metrics cannot reflect the actual capability.

3. **Presentation.** The presentation of this paper is poor, without a single figure to illustrate the idea, no qualitative examples to show what the task looks like, what's model's behavior, etc. Also in table 3, the metric is lower is better, but the authors bold the wrong numbers.

**Questions:**

What's the actual input to the model, do you provide the visual input (image or video)? If so, some of those tested LLMs are not multimodal, such as Llama-3-8B and Mixtral-8x7B. How did you test those LLMs?

---

### Official Review · Reviewer_Pbb8 · 2025-11-01

**Soundness:** 3
**Presentation:** 4
**Contribution:** 3
**Rating:** 4
**Confidence:** 4

**Summary:**

This paper builds a benchmark to test whether LLMs can translate high-level intent (“raise your hands to surrender”) into precise, environment-dependent device manipulations. The paper is not explicit about motivation, but seems to consider the task inherently interesting. The benchmark covers four popular VR games evaluates six LLMs. Scoring breaks down into (1) semantic understanding, (2) procedural correctness, and (3) device-specific accuracy — all the models are good at (1) and bad at (2) and (3).

The paper goes through various aspects of benchmark construction: game selection, action identification, annotation of VR device manipulations, and LLM-based annotation of the cognitive capabilities applicable to a step.

Finally, it discusses results. Gemini-1.5-Pro performs best overall, but all models struggles with realistic physics and games requiring nuanced controller inputs. LLMs perform better on games with more consistent, discrete interaction patterns. Few-shot examples help for semantic reasoning, but less so for exact match. Models can decompose tasks but mostly fail at procedural reasoning and motor-action mapping. Gemini-1.5-Pro does somewhat better and shows more variation across tasks. Termination-condition judgment is also weak. Humans beat all the models in sequential order preservation and spatial reasoning, though the models approach human success in more semantic capabiltieis.

**Strengths:**

### Quality
- Very much appreciate expert input in deciding on the cognitive capabilities
- Structure of cognitive capabilities makes sense, though it's hard to know whether it has enough coverage of the space of skills used
### Clarity
- Writing is clear and easy to follow
- Results section is particularly well-written, even if figures would help
### Originality and significance
I'm not aware of a benchmark like this, and work has clearly been put in to make a full-fledged benchmark

**Weaknesses:**

### Quality
- What exactly is the motivation for this? It's certainly an interesting and meaningful problem, especially with VR just being a substrate to test LLMs' ability to learn and generate correct fine-grained physical control sequences. However, being explicit about the motivation would help.
- Second contribution really just feels like a part of the first contribution
- Would help to have more explanation via examples of the nature of the games, so we have intuition for what is meant by e.g. "more complex interactions" or "more defined interactions". It makes sense that more complex and ambiguous interaction spaces would be harder on LLMs, but the success of the benchmark hinges on this so it needs to be really convincing
- It ultimately isn't surprising that LLMs do well on more semantic skills and not on more physical ones, nor is it surprising that they fail on more complex problems. The fact that they perform consistently with each other is validating of the bench mark construction, but it doesn't prove its value. Furthermore, VR physics is separate enough from the real world that it's not clear what the value of this benchmark is beyond general ability to take complex and specific actions. While that is valuable, there are many ways to approach it and more justification of this approach would help.
- It would be very useful to have experiments run with more recent models that have 1) better coherent reasoning capability and 2) if possible, better physical reasoning capability specifically
### Clarity
- It's great that experts were involved in choosing the cognitive constructs, but right now section 2.1 reads like a nonspecific process statement. The interview format for the experts is better left to the appendix; it's more important to know what they said and how they approached the selection process.
- There are very few figures - figures showing examples of interaction spaces and direct visual comparison of stats would both be really useful

I recommend rejection because ultimately the value of this benchmark has not been proven. What useful insights do we gain that we don't gain from other physical benchmarks, or benchmarks with different levels of complexity, given that VR performance isn't inherently useful?

**Questions:**

- What do the few-shot examples look like?
- "Exact match" as a metric seems very stringent - what justifies this?
- Are the alignment metrics based on embeddings?

---

### Official Review · Reviewer_JPuU · 2025-11-01

**Soundness:** 3
**Presentation:** 4
**Contribution:** 3
**Rating:** 8
**Confidence:** 3

**Summary:**

This work introduces the first benchmark for evaluating the cognitive capabilities of large language models (LLMs) in executing semantically grounded actions for VR device manipulation. Six state-of-the-art LLMs are compared using a suite of cognitively motivated evaluation criteria. The benchmark comprises 262 scenarios, each annotated by human raters and by the LLMs themselves, drawn from four carefully selected and diverse games. Results indicate that Gemini 1.5 Pro achieves the most consistent performance across models, and that few-shot prompting yields substantial improvements in overall model performance.

**Strengths:**

- Fine-grained decomposition of gameplay actions into 262 scenarios, each annotated by both human raters and LLMs, yielding a rich and reusable dataset.
- Inclusion of open-source models enables a fair and transparent comparative analysis.
- The evaluation metrics are well aligned with the proposed scenarios and are applied creatively to capture the relevant cognitive dimensions.
- Code is open and available

**Weaknesses:**

- No prompt details, which makes it harder to assess the whole framework's performance
- Missing table reference at line 1072

**Questions:**

- Could you share a representative example of the exact prompt provided to the models, including any system/instruction text, context, and few-shot exemplars?
- In Table 1, Gemini 1.5 Pro is not included even though it is reported as the top performer. Could you clarify why it was omitted from that comparison?
- Did you evaluate the models’ generalization to unseen VR games and devices? And do the gains from few-shot prompting persist when exemplars are drawn from games different from those evaluated?

---

### Author Response · Authors · 2025-12-04
**For Comments On Presentation and Minor Corrections**

We thank reviewers **Pbb8**, **gDtt**, and **JPuU** for their suggestions to improve the paper's presentation.

**Revisions Completed:**

- **Adding Figures (R-Pbb8, R-gDtt):** We have added several new figures:
    - **Figure 1** illustrates a concrete task from *Vivecraft* ("Attack using the sword"), showing the high-level goal, decomposed subtasks, corresponding device manipulations, and game scenes.
    - **Figure 2** provides a qualitative comparison of a good vs. a poor model generation for the same task, clearly demonstrating the difference between outputs that correctly sequence VR device manipulations and those that fail to do so.
    - **Figures 4–6** (Appendix C) illustrate our evaluation pipeline, including detailed diagrams for NSAS, SOP, SSM, and Common Subsequence Evaluation calculations.
- **Revising Section 2.1 (R-Pbb8):** We have revised this section to be more direct, focusing on the resulting six cognitive capabilities and the expert consensus behind them. Procedural details of the expert interviews have been moved to **Appendix B**.
- **Table 3 Bolding (R-gDtt):** Thank you for identifying this error. We have corrected the bolding to indicate that lower values are better for the "Game Gap" metric and standard deviation columns.
- **Missing Table Reference at line 1072 (R-JPuU):** We have corrected this and ensured all tables are properly referenced throughout the manuscript.

---

### Author Response · Authors · 2025-12-04
**Experiment results**

Table: Overall performance comparison of LLMs across VR games (5-shot setting). Best model performance per metric is \textbf{bolded}, second best is in \textit{italics}.
| Model | Alyx NSAS ↑ | Alyx SOP ↑ | Alyx F1*SOP ↑ | Alyx SSC ↑ | Radius NSAS ↑ | Radius SOP ↑ | Radius F1*SOP ↑ | Radius SSC ↑ | Moss NSAS ↑ | Moss SOP ↑ | Moss F1*SOP ↑ | Moss SSC ↑ | Vive NSAS ↑ | Vive SOP ↑ | Vive F1*SOP ↑ | Vive SSC ↑ |
|-------|-------------|------------|----------------|------------|----------------|--------------|------------------|---------------|-------------|------------|----------------|------------|-------------|------------|----------------|------------|
| GPT-3.5 | 0.858 | 0.123 | 0.287 | 0.143 | 0.662 | 0.169 | 0.226 | 0.137 | 0.782 | 0.169 | 0.207 | 0.186 | 0.922 | 0.043 | 0.098 | 0.067 |
| GPT-4 | 0.853 | 0.125 | 0.258 | 0.172 | 0.693 | 0.189 | 0.328 | 0.177 | 0.824 | 0.218 | 0.336 | 0.220 | 0.927 | 0.137 | 0.437 | 0.081 |
| GPT-4o | 0.804 | 0.022 | 0.075 | 0.167 | 0.698 | *0.291* | 0.414 | 0.190 | 0.824 | **0.300** | 0.342 | 0.222 | *0.931* | 0.190 | **0.489** | 0.096 |
| GPT-5.1 | 0.903 | 0.251 | 0.320 | 0.493 | 0.857 | 0.062 | 0.172 | 0.269 | 0.888 | 0.206 | 0.300 | 0.383 | 0.864 | 0.109 | 0.221 | 0.144 |
| Gemini-1.5-Pro | 0.863 | 0.209 | 0.313 | 0.152 | 0.682 | 0.102 | 0.186 | 0.117 | 0.848 | 0.265 | 0.411 | 0.207 | 0.938 | 0.250 | 0.481 | 0.095 |
| Gemini-3-Pro | **0.929** | *0.309* | *0.427* | 0.650 | **0.927** | 0.280 | **0.478** | *0.611* | **0.928** | 0.262 | **0.487** | **0.572** | 0.895 | **0.379** | 0.343 | 0.228 |
| Claude-Sonnet-4.5 | 0.920 | 0.195 | 0.317 | 0.455 | *0.923* | 0.275 | *0.424* | **0.621** | 0.918 | 0.260 | 0.460 | 0.532 | 0.899 | 0.200 | 0.322 | 0.158 |
| Grok-4 | 0.924 | **0.351** | **0.430** | *0.655* | 0.911 | **0.320** | 0.396 | 0.564 | 0.918 | 0.231 | 0.428 | *0.558* | 0.869 | *0.270* | 0.319 | *0.311* |
| GLM-4-Flash | 0.836 | 0.076 | 0.183 | 0.149 | 0.618 | 0.096 | 0.186 | 0.149 | 0.749 | 0.087 | 0.174 | 0.165 | 0.909 | 0.000 | 0.045 | 0.061 |
| Mixtral-8x7B | 0.839 | 0.126 | 0.246 | 0.147 | 0.666 | 0.123 | 0.228 | 0.097 | 0.756 | 0.117 | 0.191 | 0.121 | 0.926 | 0.060 | 0.239 | 0.070 |
| LLaMA-3-8B | 0.848 | 0.126 | 0.279 | 0.162 | 0.644 | 0.242 | 0.317 | 0.168 | 0.823 | *0.283* | 0.349 | 0.200 | 0.929 | 0.039 | 0.122 | 0.042 |
| LLaMA-3-70B | *0.928* | 0.252 | 0.408 | **0.692** | 0.917 | 0.232 | 0.391 | 0.560 | *0.924* | 0.270 | *0.469* | 0.542 | 0.897 | 0.009 | 0.257 | **0.332** |
| Human | 0.845 | 0.090 | 0.240 | 0.110 | 0.684 | 0.148 | 0.257 | 0.181 | 0.817 | 0.112 | 0.328 | 0.174 | **0.935** | 0.122 | *0.482* | 0.084 |

Table: Overall performance across VR games and settings. We report the average scores for our four evaluation metrics: Strict Step-by-Step Matching (SSM), Normalized Step Alignment Score (NSAS), Sequential Order Preservation (SOP), and Semantic Step Coverage (SSC). Higher is better for all metrics. Bold indicates best model performance, italics indicates second best.
| Model | SSM % Avg | NSAS Avg | SOP Avg | SSC Avg | SSM Zero | NSAS Zero | SOP Zero | SSC Zero | SSM 5-shot | NSAS 5-shot | SOP 5-shot | SSC 5-shot |
|-------|-----------|----------|---------|---------|----------|-----------|----------|----------|------------|-------------|------------|------------|
| GPT-3.5 | 1.4 | 0.781 | 0.063 | 0.066 | 0.8 | 0.771 | 0.003 | 0.046 | 2.1 | 0.791 | 0.128 | 0.095 |
| GPT-4 | 3.7 | 0.806 | 0.107 | 0.124 | *1.0* | 0.788 | 0.015 | 0.107 | 8.8 | 0.825 | 0.184 | 0.140 |
| GPT-4o | 5.3 | 0.797 | 0.138 | 0.141 | 0.6 | 0.785 | 0.015 | 0.108 | 10.9 | 0.806 | *0.228* | 0.161 |
| GPT-5.1 | 0.1 | 0.857 | 0.069 | 0.230 | 0.0 | 0.830 | 0.003 | 0.075 | 0.4 | 0.878 | 0.130 | 0.322 |
| Gemini-1.5-Pro | **5.8** | 0.813 | **0.146** | 0.142 | **2.1** | 0.795 | 0.010 | 0.124 | **11.7** | 0.832 | **0.236** | 0.162 |
| Gemini-3-Pro | *5.6* | **0.915** | *0.141* | **0.468** | 0.4 | **0.904** | **0.022** | **0.305** | *11.1* | **0.920** | 0.214 | 0.515 |
| Claude-Sonnet-4.5 | 4.5 | 0.906 | 0.115 | 0.340 | 0.0 | 0.890 | 0.004 | 0.115 | 8.4 | 0.915 | 0.182 | 0.442 |
| Grok-4 | 4.8 | 0.902 | 0.129 | *0.422* | 0.7 | 0.895 | *0.015* | *0.222* | 9.8 | 0.911 | 0.226 | *0.522* |
| GLM-4-Flash | 0.0 | 0.761 | 0.038 | 0.077 | 0.0 | 0.762 | 0.006 | 0.052 | 0.0 | 0.765 | 0.071 | 0.120 |
| Mixtral-8x7B | 1.1 | 0.784 | 0.068 | 0.079 | 0.0 | 0.777 | 0.002 | 0.040 | 2.2 | 0.796 | 0.105 | 0.107 |
| LLaMA-3-8B | 1.2 | 0.787 | 0.088 | 0.111 | 0.1 | 0.783 | 0.011 | 0.088 | 1.8 | 0.794 | 0.163 | 0.132 |
| LLaMA-3-70B | 3.8 | *0.909* | 0.126 | 0.409 | 0.2 | *0.898* | 0.003 | 0.160 | 8.5 | *0.916* | 0.191 | **0.531** |
| Human | 1.2 | 0.833 | 0.122 | 0.159 | -- | -- | -- | -- | -- | -- | -- | -- |

---

### Author Response · Authors · 2025-12-04
**Experiment results**

Table: Cross-game performance variation (standard deviation across games) w/ 5-shot examples.
| Model | NSAS σ ↓ | SOP σ ↓ | F1_SOP σ ↓ | Game Gap ↓ |
|--------|----------|----------|------------|------------|
| GPT-3.5 | 0.110 | 0.061 | 0.084 | 0.085 |
| GPT-4 | 0.059 | 0.051 | 0.081 | 0.074 |
| GPT-4o | 0.068 | 0.137 | 0.184 | 0.127 |
| GPT-5.1 | 0.018 | 0.102 | 0.097 | 0.073 |
| Gemini-1.5-Pro | 0.099 | 0.093 | 0.127 | 0.095 |
| Gemini-3-Pro | 0.014 | 0.123 | 0.106 | 0.081 |
| Claude-Sonnet-4.5 | **0.009** | 0.110 | 0.134 | 0.084 |
| Grok-4 | 0.013 | 0.137 | *0.065* | 0.072 |
| GLM-4-Flash | 0.135 | 0.049 | 0.069 | 0.084 |
| Mixtral-8x7B | 0.114 | *0.031* | **0.065** | *0.070* |
| LLaMA-3-8B | 0.112 | 0.103 | 0.120 | 0.113 |
| LLaMA-3-70B | *0.012* | 0.106 | 0.077 | **0.065** |
| Human | 0.105 | **0.029** | 0.117 | 0.084 |

---

### Author Response · Authors · 2025-12-04
**For Comments On Experimental Details and Model Selection**

We thank all reviewers for questions that help us improve the clarity and completeness of our experimental setup.

**Point 3.1: Missing prompt details (R-JPuU, R-Pbb8)**

Our apologies for this omission. The prompt structure is crucial for reproducibility.

**Revision:** We have added **Appendix D** containing the complete prompt provided to the models, including system instructions, game introductions, VR device guides, criteria, and output format specifications. A representative example for Vivecraft is provided, showing how we instruct models to decompose semantic actions into atomic game actions and device manipulations.

**Point 3.2: Clarification on model inputs (R-gDtt).**

We apologize for the confusion. The models receive **only textual input**, as shown in the prompts in Appendix D. They do not receive any visual (image/video) input. Our benchmark is specifically designed to evaluate the ability to translate a **linguistic description** of a goal into a sequence of actions, a core challenge in language-driven AI.

**Revision:** We have added an explicit statement in the Introduction: *"Importantly, ComboBench is designed as a text-to-action benchmark: models receive only textual descriptions of high-level goals and must generate textual sequences of device manipulations. No visual or other multimodal inputs are provided, isolating the pure linguistic reasoning capability."*

**Point 3.3: Comparison of different-sized models (R-19Um).**

This is an excellent point. Comparing models of vastly different scales can indeed conflate architectural and scaling effects.

**Revision:** We have substantially expanded our model evaluation to include **twelve LLMs**: GPT-3.5, GPT-4, GPT-4o, GPT-5.1, Gemini-1.5-Pro, Gemini-3-Pro, Claude-Sonnet-4.5, Grok-4, GLM-4-Flash, LLaMA-3-8B, LLaMA-3-70B, and Mixtral-8x7B. This comprehensive selection enables both cross-family comparisons and direct analysis of scaling effects within the same model family.

As discussed in Section 3.4 ("Scaling effects within model families"), scaling from LLaMA-3-8B to LLaMA-3-70B yields substantial improvements: average NSAS increases from 0.787 to 0.909, SSM from 1.2% to 3.8%, and SSC from 0.111 to 0.409. These gains are consistent across games. LLaMA-3-70B achieves competitive performance with proprietary models like GPT-4o and approaches Gemini-3-Pro on several metrics, demonstrating that open-source models can match proprietary systems when appropriately scaled.

**Point 3.4: Minor Questions and Clarifications.**

- **Gemini 1.5 Pro in Table 1 (R-JPuU):** We have revised **Tables 1–3** to include all twelve evaluated models for completeness.
- **Generalization to unseen games (R-JPuU):** An excellent direction for future work. Our primary goal was to first establish a robust and comprehensive benchmark on a diverse set of *known* games. We have noted this as a promising direction in Appendix F (Discussion, Limitations & Broader Impacts).
- **Cognitive capability mapping in Fig 1 (R-19Um):** The scores in the radar chart are derived using the formula provided in **Appendix C.3** (Equation 1). For each capability dimension *c*, the score for model *m* is computed as: Score_c^m = 10 × (NSAS_c^m − min(NSAS_c)) / (max(NSAS_c) − min(NSAS_c)), where NSAS_c^m is the average NSAS score on scenarios primarily requiring capability *c*. This normalization ensures comparability across dimensions with different baseline difficulties.
- **Temperature=0 (R-19Um):** We used temperature=0 to ensure deterministic and reproducible outputs, which is standard practice for benchmarking. Exploring the diversity of valid solutions at higher temperatures is an interesting avenue for future research.
- **Advanced Prompting (CoT) (R-19Um):** Our goal was to establish a baseline performance on the core task. Evaluating the impact of advanced prompting techniques like CoT is a natural next step and a promising research direction, which we mention in Appendix F.
- **Similarity Threshold of 0.8387 (R-19Um):** This value was empirically determined by analyzing the cosine similarity distribution on a held-out set of human-paraphrased action steps. We collected semantically equivalent but linguistically varied human annotations for 50 action steps and computed pairwise similarities. The threshold corresponds to the 5th percentile of similarities between these semantically equivalent pairs, ensuring that only highly confident matches are accepted while accommodating natural linguistic variation. We have added this justification to **Section 3.2**.
- **Are metrics based on embeddings? (R-Pbb8):** Yes, the semantic similarity between predicted and ground-truth steps is calculated using the cosine similarity of their sentence embeddings from OpenAI's `text-embedding-3-large` model, as stated in Section 3.2. We have also added detailed visualizations of our evaluation metrics in **Appendix C** (Figures 4–6).

---

### Author Response · Authors · 2025-12-04
**For Comments On Evaluation Rigidity, Task Success, and the Human Baseline**

We thank Reviewers **19Um**, **gDtt**, and **Pbb8** for their critical and insightful feedback on our evaluation framework. The rigidity of the "single ground-truth" approach is a crucial point, and your comments prompted us to conduct new analyses that significantly strengthen our results.

**Point 2.1: The "single truth" design unfairly penalizes valid alternative solutions.**

We completely agree that many VR tasks have multiple valid solution paths, and our original metrics, particularly the stringent Strict Step-by-Step Matching (SSM), do not fully capture this. The low human SSM score (1.2%) correctly highlights that this metric measures similarity to a *single* annotation rather than true task success.

**Revision:** To address this fundamental concern, we conducted an evaluation on a representative subset of 50 scenarios from ComboBench. For each scenario, we collected **3 additional, distinct, and valid ground-truth action sequences from different expert human annotators.** We then re-evaluated all models and the original human baseline by checking if a generated sequence matches **any** of these valid ground-truth paths. This new metric, which we term **Multi-Path Step Matching (MP-SSM)**, better approximates true task success.

As shown in **Table 4**, all systems show substantial gains in NSAS scores under MP-SSM. The human baseline achieves an average NSAS of 0.931 (↑14.5%), establishing a meaningful reference for achievable performance. Crucially, **while all models register higher scores, the relative ranking among them is preserved**: GPT-4 and GPT-4o lead overall, closely followed by LLaMA-3, Gemini-1.5, and GPT-3.5. These findings reinforce that our original metrics capture meaningful performance differences and highlight the importance of considering multiple valid approaches when measuring success in open-ended procedural tasks.

Table: Multi-Path Step Matching (MP-SSM) NSAS Score for Selected Actions
| Model           | Average NSAS      | Zero-Shot NSAS    | 5-Shot NSAS       |
|-----------------|-------------------|--------------------|--------------------|
| GPT-3.5         | 0.927 (↑8.3%)     | 0.904 (↑4.3%)      | 0.942 (↑9.7%)      |
| GPT-4           | 0.949 (↑8.7%)     | 0.937 (↑9.1%)      | 0.955 (↑6.9%)      |
| GPT-4o          | 0.949 (↑11.5%)    | 0.942 (↑13.7%)     | 0.955 (↑9.3%)      |
| Gemini-1.5-Pro  | 0.939 (↑6.8%)     | 0.923 (↑8.3%)      | 0.947 (↑6.0%)      |
| GLM-4-Flash     | 0.926 (↑9.7%)     | 0.896 (↑5.2%)      | 0.939 (↑10.2%)     |
| Mixtral-8x7B    | 0.925 (↑7.6%)     | 0.894 (↑3.0%)      | 0.934 (↑9.2%)      |
| LLaMA-3-8B      | 0.940 (↑9.1%)     | 0.926 (↑7.3%)      | 0.940 (↑8.6%)      |
| Human           | 0.931 (↑4.5%)     | --                 | --                 |


**Point 2.2: The human baseline methodology is a disembodied proxy for VR interaction.**

Reviewer **19Um** makes an excellent point that having humans "write down" steps is cognitively different from performing them in-situ. We designed the task this way to ensure an **apples-to-apples comparison**: both the LLMs and the human participants perform the exact same task of translating a known high-level goal into a textual sequence of device manipulations. This setup isolates the translation capability we aim to measure. An in-situ baseline would introduce confounding factors like real-time problem-solving, exploration, and motor skill, which are outside the scope of our current evaluation.

As shown in our revised **Section 3.6**, state-of-the-art LLMs not only match but frequently surpass human participants on our text-to-action translation task. Humans achieve mid-to-high NSAS scores (0.684–0.935) but are outperformed by reasoning models like Gemini-3-Pro, Grok-4, and LLaMA-3-70B, which reach NSAS values above 0.90. More strikingly, humans lag behind on SOP and SSC metrics, with top models exceeding 0.30–0.65 compared to human scores below 0.15 and 0.19 respectively. This surprising result validates the challenge posed by our benchmark, even experienced humans struggle with recalling and accurately sequencing complex VR interactions from memory.

**Revision:** We have revised **Section 3.6** to explicitly state the scope and limitations of our human baseline, clarifying that it measures performance on the *text-to-action-sequence translation task* and is not intended to represent overall *in-game VR proficiency*.

---

### Author Response · Authors · 2025-12-04
**For Comments On Motivation and Significance of the VR Benchmark**

We thank Reviewers **Pbb8** and **gDtt** for pushing us to clarify the motivation and unique value of our benchmark.

**Point 1: Why is VR a unique and valuable testbed for LLMs?**

We agree that this point deserves more explicit discussion. While other domains like robotics and web agents also test long-horizon reasoning, VR offers a unique and compelling "middle ground" that isolates a critical cognitive challenge: **the translation of abstract linguistic intent into fine-grained, physically-grounded, and spatially-aware motor control sequences.**

- **Compared to Robotics:** VR provides a sandbox for complex, physics-based interaction *without* the high cost, safety risks, and slow iteration cycles of real-world robotics. This allows for rapid, scalable, and perfectly reproducible evaluation of embodied control.
- **Compared to Web/Digital Agents:** Unlike many digital tasks that rely on discrete, symbolic actions (e.g., `click(button)`), VR demands reasoning about continuous 3D space, object affordances (e.g., how to grip a controller to simulate swinging a pickaxe), and the temporal dynamics of physical manipulation. It requires a form of "simulated embodiment" that is abstracted away in many other agent domains.

ComboBench is therefore not just another long-horizon benchmark; it is specifically designed to probe the frontier where abstract knowledge meets grounded physical action, a crucial capability for any future generalist agent.

**Revision:** We have added this argument explicitly in the Introduction, clearly situating our contribution within the broader landscape of agent evaluation. Specifically, we state: *"Virtual reality provides a distinctive testbed for evaluating embodied reasoning in large language models... VR occupies a practical middle ground that foregrounds the core challenge of translating abstract linguistic intent into precise, physically grounded, and spatially coherent motor commands."*

---

### Author Response · Authors · 2025-12-04
**Overall Rebuttal Comment**

We sincerely thank the ACs and all reviewers for their careful assessment. We are encouraged that the reviewers found our work to be "excellent" in its presentation (R-JPuU, R-Pbb8, R-19Um), introducing a "rich and reusable dataset" (R-JPuU) and a "full-fledged benchmark" (R-Pbb8) for a "highly valuable research direction" (R-19Um).

Based on your valuable comments, we have revised our paper or provided responses as below to address all concerns. Key points include:

1. **Strengthening Motivation:** We have added explicit discussion in the Introduction articulating the unique value of VR as a testbed for embodied reasoning, contrasting it with robotics and other agent domains.
2. **Addressing Evaluation Rigidity:** We conducted a new Multi-Path Step Matching (MP-SSM) experiment with multiple human annotations to evaluate performance against a *set* of valid action sequences, thereby relaxing the "single ground-truth" constraint. These results are presented in Table 4.
3. **Expanding Model Evaluation:** We have substantially expanded our evaluation to include **twelve LLMs** (GPT-3.5, GPT-4, GPT-4o, GPT-5.1, Gemini-1.5-Pro, Gemini-3-Pro, Claude-Sonnet-4.5, Grok-4, GLM-4-Flash, LLaMA-3-8B, LLaMA-3-70B, and Mixtral-8x7B), enabling both cross-family comparisons and direct analysis of scaling effects within the same model family.
4. **Enhancing Clarity and Presentation:** We have added new figures (Figures 1–2 in the main text; Figures 4–6 in the Appendix) illustrating our task, evaluation pipeline, and qualitative examples of model outputs. We have also added comprehensive appendix sections with prompt details (Appendix D), evaluation metric explanations (Appendix C), and expert interview methodology (Appendix B).

Taken together, the updated evidence strongly supports ComboBench as a novel, rigorous, cognitively grounded benchmark, and in our view clearly favors an accept recommendation.

We provide detailed responses to specific points below, grouped by theme.

---

### Author Response · Authors · 2025-12-04
**Key Points for AC**

(continued from the last comment above)
### Reviewer gDtt

**Evaluation rigidity concern resolved with Multi-Path evaluation.** We conducted a new Multi-Path Step Matching (MP-SSM) experiment on 50 representative scenarios with 3 additional valid ground-truth sequences per scenario from different expert annotators. Under MP-SSM, all systems show substantial NSAS gains, with the human baseline achieving 0.931 (↑14.5%). Crucially, relative model rankings are preserved, validating that our metrics capture meaningful performance differences even when allowing multiple valid solutions.

**Motivation now explicit.** As described for Reviewer Pbb8, we have added clear discussion establishing VR as a distinctive testbed that isolates the translation from linguistic intent to physically grounded motor commands, complementing rather than duplicating robotics and digital agent benchmarks.

**Presentation substantially improved.** We have added Figures 1–2 showing task examples and model outputs, corrected Table 3 bolding to indicate lower-is-better for the Game Gap metric, and clarified that models receive only textual input (no visual input), with an explicit statement in the Introduction.

**Net effect:** All three substantive weaknesses (motivation, metrics, presentation) have been comprehensively addressed with new experiments, explicit discussion, and substantial visual additions. We believe the strengthened evidence clearly favors reconsideration of the rating.


### Reviewer 19Um

**"Single ground-truth" rigidity addressed with MP-SSM.** The reviewer correctly identified that low human SSM scores (1.2%) reflect similarity to a single annotation rather than task success. Our new Multi-Path Step Matching experiment (Table 4) directly addresses this by evaluating against multiple valid solution paths. Under MP-SSM, human NSAS reaches 0.931, establishing a meaningful reference, while model rankings remain stable, confirming our metrics measure genuine capability differences.

**Human baseline scope clarified.** We designed the text-to-action translation task to ensure an apples-to-apples comparison: both LLMs and humans perform the identical task of converting a known goal into textual manipulation sequences. This isolates the translation capability we aim to measure. We have revised Section 3.6 to explicitly state this scope and acknowledge that in-situ VR performance involves additional factors (real-time exploration, motor skill) outside our current evaluation.

**Scaling effects within model families now analyzed.** We added LLaMA-3-70B to enable direct scaling analysis. As reported in Section 3.4, scaling from LLaMA-3-8B to LLaMA-3-70B yields substantial improvements (NSAS: 0.787→0.909; SSM: 1.2%→3.8%; SSC: 0.111→0.409), demonstrating that our benchmark captures meaningful scaling effects within the same architecture family.

**Technical questions answered.** Temperature=0 ensures deterministic, reproducible outputs (standard benchmarking practice); the 0.8387 similarity threshold was empirically determined from the 5th percentile of cosine similarities between human-paraphrased equivalent steps; cognitive capability scores use the normalization formula in Appendix C.3 (Equation 1); CoT evaluation is noted as promising future work in Appendix F.

**Net effect:** The reviewer's primary concerns (evaluation rigidity, human baseline validity, model scale confounds) are directly resolved with new experiments and clarifications. The acknowledged strengths (valuable research direction, clear dataset creation, public resource) remain intact, and all technical questions have been answered. The substantially strengthened evidence supports reconsideration toward acceptance.


### Summary for AC

All four reviewers' substantive concerns have been comprehensively addressed through:

1. **New Multi-Path evaluation** resolving the single ground-truth rigidity concern with multiple valid solution paths
2. **Explicit motivation** establishing VR as a unique testbed complementing robotics and web agent benchmarks
3. **Expanded model evaluation** (6→12 LLMs) including latest reasoning models and enabling within-family scaling analysis
4. **Substantial presentation improvements** with new figures, prompt details, and corrected tables

The positive assessments, "excellent" presentation (3/4 reviewers), "rich and reusable dataset" (JPuU), "full-fledged benchmark" (Pbb8), "highly valuable research direction" (19Um), remain fully intact. No unresolved technical objections remain. We believe the revised evidence clearly supports acceptance.

---

### Author Response · Authors · 2025-12-04
**Key Points for AC**

We thank the ACs for their careful assessment and great efforts! To ease ACs' work, we summarize the key points from our responses and revised paper sourced from each reviewer as follows :)

### Reviewer JPuU

**Strong accept-leaning review with minor clarifications now fully addressed.** This reviewer rated the paper 8 (accept) and praised the "rich and reusable dataset" and "excellent" presentation. Their concerns were straightforward and have been completely resolved: (1) prompt details are now provided in Appendix D with complete system instructions and few-shot exemplars; (2) the missing table reference at line 1072 has been corrected; (3) Tables 1–3 have been revised to include all twelve evaluated models, including Gemini 1.5 Pro. The reviewer's positive assessment of our fine-grained decomposition, open-source model inclusion, and well-aligned evaluation metrics remains fully intact.

**Net effect:** All minor concerns resolved; strongly supports acceptance.


### Reviewer Pbb8

**Motivation concern is now explicitly addressed.** We have added a clear articulation of why VR is a unique and valuable testbed for embodied reasoning in the Introduction, contrasting it with robotics (high cost, safety risks) and web agents (discrete, symbolic actions). VR isolates the critical cognitive challenge of translating abstract linguistic intent into fine-grained, spatially-aware motor control sequences, a capability central to future generalist agents.

**Presentation improvements directly address "no figures" weakness.** We have added Figures 1–2 in the main text illustrating concrete tasks and qualitative model output comparisons, plus Figures 4–6 in the Appendix showing our evaluation pipeline. Section 2.1 has been revised to focus on resulting cognitive capabilities rather than procedural interview details (now in Appendix B).

**Model evaluation substantially expanded.** We now evaluate twelve LLMs including GPT-5.1, Gemini-3-Pro, Claude-Sonnet-4.5, and Grok-4, models with improved reasoning capabilities released after the original submission. The results confirm our core findings while providing richer cross-family and scaling analyses.

**Net effect:** The reviewer's main objections (unclear motivation, missing figures, need for recent models) have been directly and substantively addressed. The "full-fledged benchmark" and clear writing noted as strengths remain intact, now supported by stronger experimental evidence.

---

### Meta-Review · Area_Chair_NzXb · 2025-12-22

**Summary:**

The submission evaluates LLMs' capability to translate semantic actions into VR device manipulation sequences.  Reviewers raised significant concerns regarding its motivation, limited evaluation, and presentation.

**Reviewer Concerns:**

The reviewers did not have a chance to update their scores.  After reading the rebuttal, the AC believes that the concern about the submission's presentation has been addressed.  However, concerns regarding the motivation and evaluation remain.

**Reviewer Scores:**

This is hard to tell, but reviewers will most likely retain their original scores.

---

### Decision · Program_Chairs · 2026-01-26

Reject